# Maximizing Information in Domain-Invariant Representation Improves Transfer Learning

## Abstract

The most effective domain adaptation (DA) technique involves the decomposition of data representation into a domain-independent representation (DIRep) and a domain-dependent representation (DDRep). A classifier is trained by using the DIRep on the labeled source images. Since the DIRep is domain invariant, the classifier can be "transferred" to make predictions for the target domain with no (or few) labels. However, information useful for classification in the target domain can "hide" in the DDRep. Current DA algorithms, such as Domain-Separation Networks (DSN), do not adequately address this issue. DSN's weak constraint to enforce the orthogonality of DIRep and DDRep allows this hiding effect and can result in poor performance. To address this shortcoming, we develop a new algorithm wherein a stronger constraint is imposed to minimize the information content in DDRep to create a DIRep that retains relevant information about the target labels and, in turn, results in a better invariant representation. By using synthetic datasets, we show explicitly that depending on the initialization, DSN, with its weaker constraint, can lead to sub-optimal solutions with poorer DA performance. In contrast, our algorithm is robust against such perturbations. We demonstrate the equal-or-better performance of our approach against DSN and other recent DA methods by using several standard benchmark image datasets. We further highlight the compatibility of our algorithm with pre-trained models for classifying real-world images and showcase its adaptability and versatility through its application in network intrusion detection.

## 1 Introduction

Labeling data for machine learning can be a difficult and time-consuming process. If we have a set of labels for data drawn from a source domain, it is desirable to use the source data and labels to aid in classifying data from a similar but different target domain with no (or few) labels. Transferring the ability to classify data from one domain to another is called *Domain Adaptation* (DA).

Humans looking at pictures of huskies and wolves in the wild often notice the background of the animal to aid in classification. For example, images of wolves often depict the animals in wild settings, which are rarely associated with images of huskies. However, when the source domain comprises images of wolves and huskies in their natural habitats, and the target domain consists of them in veterinary clinics, the contextual cues present in the source domain are no longer available in the target domain. As a result, the contextual clues can be thought of as domain-specific information or "spurious information" since the physical characteristics of either animal provide the necessary clues to distinguish them. The loss of domain-specific information poses significant challenges to accurate classification across domains, as neural networks can become overly reliant on such information to infer labels during training (Zewe, 2021). Our goal is to ensure that neural networks use domain-independent information to produce effective DA.

Our general intuition, largely consistent with previous work (Stojanov et al., 2021; Bousmalis et al., 2016), is that effective DA can occur if two requirements are met:

1. A representation of the input can be formed that is independent of the domain, which we call a *Domain-Independent Representation (DIRep)*.

2. The DIRep contains all the relevant information for the classification in the target domain.

A well-known approach to address requirement 1 is using adversarial techniques such as generative adversarial networks (GANs) (Ganin et al., 2016; Singla et al., 2020; Tzeng et al., 2017). These adversarial techniques ensure that from the DIRep, one cannot determine which domain the original data came from. However, the GAN alone does not guarantee that the learned DIRep will have any relevant information for predicting the label in the target domain (Stojanov et al., 2021).

To satisfy requirement 2, one strategy is to put all the data information (in both domains) into the representation by using an autoencoder approach. However, this can not be done with the DIRep alone, as it is not supposed to contain domain-dependent information. To circumvent this problem, a *Domain-Dependent Representation (DDRep)* can be introduced as in previous methods such as Domain-Separation-Networks (DSN) (Bousmalis et al., 2016). The data is represented by the DIRep and DDRep, which can be used together to reconstruct the data in the autoencoder. We adopt the same approach in our study but with a different way of decomposing the DDRep and DIRep from that used in DSN.

One of the main challenges in DA is determining what goes into the DIRep and what goes into the DDRep. Our approach, called MaxDIRep, uses a KL divergence constraint between the DDRep and a standard normal distribution to ensure that the bare minimum of information goes into the DDRep. We thus ensure that the DIRep contains as much relevant information as possible, consistent with requirement 1, which is achieved using adversarial techniques. The only information required in the DDRep is the information about what domain it comes from, which is not useful in classification. This differs from DSN, which only constrains DDRep to be orthogonal with DIRep. As a result, useful information for classifying the target domain may end up in the DDRep and cannot be used for classifying the target data.

The rest of our paper is structured as follows. After discussing related work in Section 2, we present details of our approach and contrast it to the closely related DANN (Ganin et al., 2016) and DSN (Bousmalis et al., 2016) algorithms in Section 3. In Section 4, we give results on a synthetic benchmark we designed to elucidate the issues impacting previous methods, an ablation study that further illustrates the advantage of our approach versus DSN, and the performance of our algorithm versus other DA methods across a set of standard image benchmark datasets. Finally we show the superior results of our approach on a non-image classification task. In Section 5, we discuss the intuitive reason for the better performance of our approach and possible future directions for further improvements.

## 2  Related work

Transfer learning is an active research area that has been covered by several survey papers (Liu et al., 2022; Zhang & Gao, 2022; Zhang, 2021; Zhuang et al., 2020; Liu et al., 2019; Wang & Deng, 2018). Here, we briefly describe previous methods focusing on those that are closely related to ours.

The domain adversarial neural network (DANN) (Ganin et al., 2016) uses three network components, namely a feature extractor, a label predictor, and a domain classifier. The generator is trained in an adversarial manner to maximize the loss of the domain classifier by reversing its gradients. The generator is trained at the same time as the label predictor to create a DIRep that contains domain-invariant features for classification. The adversarial discriminative domain adaptation (ADDA) (Tzeng et al., 2017) approach adopts similar network components with a learning process that involves multiple stages in training the three components of the model. Singla et al. (2020) has proposed a hybrid version of the DANN and ADDA where the generator is trained with the standard GAN loss function (Goodfellow et al., 2020). We refer to this as the GAN-based method (Singla et al., 2020). None of these methods (DANN, ADDA, and GAN-based) includes the auto-encoder and thus does not have a DDRep.

The Domain-Specific Adversarial Network (DSAN) (Stojanov et al., 2021) makes use of domain-specific information, but it is used by the encoding function, in addition to the input data, to infer the DIRep. In contrast, our approach learns the DIRep without incorporating the domain-specific information as the input. The closest approach to ours is the Domain-Separation-Networks (DSN) (Bousmalis et al., 2016). The key distinction between DSN and our method is the different constraints used in the decomposition of the data

representation into DIRep and DDRep. In DSN, the DDRep and DIRep have the same shape, and a linear "soft subspace orthogonality constraint between the private and shared representation of each domain" was used to ensure that the DIRep and DDRep are different. In our approach, a stronger constraint to minimize information content in DDRep is used. Details are described in Section 3.3.

Other work shows how to take advantage of more than one target (Peng et al., 2019), or more than one source domain (Pei et al., 2018; Park & Lee, 2021). Some authors have evaluated the quality of cross-domain representation disentanglement on image-to-image translation and image retrieval tasks such as the Interaction Information Auto-Encoder (IIAE) (Hwang et al., 2020). The Variational Disentanglement Network (VDN) (Wang et al., 2022) attempts to generalize from a source domain without access to a target. These methods either focus on different problem settings (Peng et al., 2019; Pei et al., 2018; Park & Lee, 2021; Wang et al., 2022) (instead of one source domain and one target domain) or utilize non-adversarial training techniques to learn domain-invariant features (Hwang et al., 2020). These works are less related to our approach and will thus not be discussed in-depth in this paper.

## 3 The MaxDIRep algorithm for domain adaptation

In this section, we describe the details of our method (MaxDIRep), which is summarized in Figure 1. To achieve an effective adaptation, our goal is to constrain the DIRep extraction to ensure it retains the "maximal" amount of information about the target labels. Specifically, MaxDIRep achieves this by enforcing the minimal information content in the DDRep during the data generation process from both DDRep and DIRep. We measure the KL divergence on DDRep with respect to a standard normal distribution (which is considered as the baseline distribution that has little information). Including this KL divergence in the overall loss function allows us

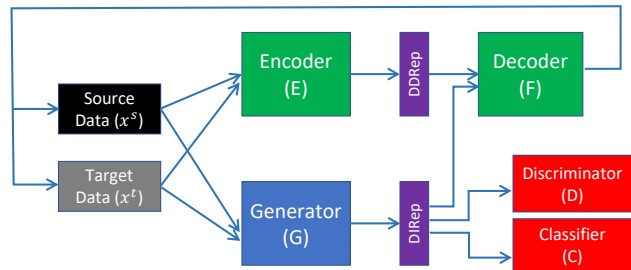

Figure 1: Architecture of MaxDIRep.

to constrain the information content in DDRep. Since DDRep corresponds to minimal information specific to each domain, this forces DIRep to retain maximal information about the target labels. DIRep is also subject to a GAN-like discriminator ensuring that the classification information is domain-invariant. Details of the MaxDIRep method are described below:

**(1) Networks.** There are five neural networks (by neural network, we mean the network architecture and all its parameters) in the algorithm: 1) $G$ is the generator; 2) $D$ is the discriminator; 3) $C$ is the classifier; 4) $E$ is the encoder; 5) $F$ is the decoder.

**(2) Inputs and outputs.** The data is given by $(x, l, d)$ where $x$ is the input; we use the notation $x^s$ and $x^t$ to respectively represent the source and target data samples, when necessary to distinguish them. $l$ is the label of sample $x$ (if any), and $d$ is the domain identity (e.g., it can be as simple as one bit of 0 for the source domain and 1 for the target domain). In the zero-shot or few-shot DA settings, $l$ is available for all source data samples, but none or only a few labels are known for the target samples. $x$ is the input given to both encoder ($E$) and generator ($G$). The DDRep and DIRep correspond to the intermediate outputs of $E$ and $G$, respectively:

$$DDRep = E(x), \quad DIRep = G(x), \tag{1}$$

which then serve as the inputs for the downstream networks decoder ($F$), discriminator ($D$), and classifier ($C$). In particular, DIRep serves as the input for $D$ and $C$, and both DIRep and DDRep serve as the inputs for $F$. The outputs of these three downstream networks are $\hat{x}$ from the decoder $F$, $\hat{d}$ from the discriminator $D$, and $\hat{l}$ from the classifier $C$:

$$\hat{x} = F(E(x), G(x)), \quad \hat{d} = D(G(x)), \quad \hat{l} = C(G(x)), \tag{2}$$

where we explicitly list the dependence of the outputs on the corresponding networks.

**(3) Loss functions.** Some measures of the differences between the predictions from the networks, i.e., $(\hat{x}, \hat{d}, \hat{l})$ and their actual values $(x, d, l)$ are used to construct the loss functions. Typically, a loss function would take two arguments: a prediction and the actual label/value. We use the name of the loss function without specifying the arguments and do so for the discriminator, generator, classification and reconstruction losses. All the loss functions with their dependence on specific neural networks are given explicitly here:

1. Classification loss: $\mathcal{L}_c = \mathcal{L}_c(\hat{l}, l) = \mathcal{L}_c(C(G(x), l))$.

2. Discriminator loss: $\mathcal{L}_d = \mathcal{L}_d(\hat{d}, d) = \mathcal{L}_d(D(G(x), d))$.

3. Generator loss: $\mathcal{L}_g = \mathcal{L}_g(\hat{d}, 1 - d) = \mathcal{L}_d(D(G(x)), 1 - d)$.

4. Reconstruction loss: $\mathcal{L}_r = \mathcal{L}_r(\hat{x}, x) = \mathcal{L}_r(F(G(x), E(x)), x)$.

5. KL loss for DDRep: $\mathcal{L}_{kl} = D_{KL}(Pr(E(x)) \parallel \mathcal{N}(0, I))$.

For the reconstruction loss $\mathcal{L}_r$, we use the $L_2$-norm. For $\mathcal{L}_d, \mathcal{L}_g, \mathcal{L}_c$, we use cross entropy. A more detailed formulation of the loss functions is given in the Appendix A.

The first four loss functions ($\mathcal{L}_c$, $\mathcal{L}_d$, $\mathcal{L}_g$, and $\mathcal{L}_r$) are similar to those used in other GAN-based algorithms such as DSN. The most important and unique feature of our algorithm is the KL divergence loss function $\mathcal{L}_{kl}(E)$ for the DDRep ($E$). $\mathcal{L}_{kl}$ is introduced to create a DDRep that has little information so that DIRep has to play a major role in the decoder when reconstructing the data, which, in turn, forces DIRep to include sufficient information for target classification.

**(4) The back-prop based learning.** The gradient descent based learning dynamics for updating the five neural networks are described by the following equations:

$$\Delta G = -\alpha_G \left( \lambda \frac{\partial \mathcal{L}_g}{\partial G} + \beta \frac{\partial \mathcal{L}_c}{\partial G} + \gamma \frac{\partial \mathcal{L}_r}{\partial G} \right), \Delta C = -\alpha_C \frac{\partial \mathcal{L}_c}{\partial C}, \quad \Delta D = -\alpha_D \frac{\partial \mathcal{L}_d}{\partial D},$$

$$\Delta E = -\alpha_E \left( \frac{\partial \mathcal{L}_{kl}}{\partial E} + \mu \frac{\partial \mathcal{L}_r}{\partial E} \right), \quad \Delta F = -\alpha_F \frac{\partial \mathcal{L}_r}{\partial F},$$

where $\alpha_{C,D,E,F,G}$ are the learning rates for different neural networks. In our experiments, we often set them to the same value, but they can be different in principle. The other hyperparameters, namely $\lambda$, $\beta$, $\gamma$, and $\mu$, are the relative weights of the loss functions. These hyperparameters are also useful to understand the different algorithms. As easily seen from the equations above, when $\gamma = 0$, the GAN-based algorithm decouples from the VAE-based constraints.

## 3.1 The explicit DDRep algorithm

From the results of the full MaxDIRep algorithm, we found that the DDRep contains a small amount of information as measured by the KL divergence, which is consistently small in all the experiments, see Table 13. Inspired by this observation, we introduce a simplified MaxDIRep algorithm without the encoder $E$ wherein the DDRep is set explicitly to be just the domain label (bit) $d$, i.e., $DDRep = d$. We call this simplified MaxDIRep algorithm the *explicit DDRep* algorithm. The motivation is that $d$ is the simplest possible domain-dependent information that could serve to filter out the domain-dependent information from the DIRep.

Besides its simplicity, the explicit DDRep algorithm is also highly interpretable. One particularly useful feature of the explicit DDRep algorithm is that it allows us to check the effect of the DDRep directly by flipping the domain bit ($d \rightarrow 1 - d$). We know that the domain bit is effective in filtering out domain-dependent information from the DIRep if the reconstructed image $\tilde{x} = F(DIRep, 1 - d)$ resembles an image from the other domain (see Section 4.1.1 for details and Figure 3a for examples of reconstructed images).

In the experiments, the explicit DDRep algorithm has the same performance as the full MaxDIRep model in some simpler cases (see Section 4.1.1). However, the full MaxDIRep model performs better in more complex

cases (Sections 4.1.2, 4.3 and 4.4). Therefore, we use the full MaxDIRep model with $\mathcal{L}_{kl}$ for all cases as it is more general except in the experiments in Section 4.1.1 where the explicit DDRep algorithm works just as well but also provides a direct interpretation of the algorithm (See Figure 3a).

## 3.2 Comparing MaxDIRep to DANN: insights from domain adaptation theory

We now give a theoretical understanding of MaxDIRep based on the domain adaptation theory established in Theorem 1 from Ben-David et al. (2010). While deriving the explicit target error bound for MaxDIRep turns out to be formidable, we provide some insights into why MaxDIRep yields better adaptability than DANN, grounded in Theorem 1. These insights will be empirically validated through experiments presented in Subsection 4.1.1.

**Theorem 1.** (Ben-David et al. (2010)). Let $\mathcal{H}$ be the hypothesis space and $\mathcal{E}_s(h)$, $\mathcal{E}_t(h)$ be the error of hypothesis $h \in \mathcal{H}$ on the source domain $X_s$ and the target domain $X_t$, respectively. Then for any classifier $h \in \mathcal{H}$, the error on the target domain is bounded by,

$$\mathcal{E}_t(h) \leq \mathcal{E}_s(h) + d_{\mathcal{H}\Delta\mathcal{H}}(X_s, X_t) + \lambda, \tag{3}$$

where $d_{\mathcal{H}\Delta\mathcal{H}}$ is the $\mathcal{H}\Delta\mathcal{H}$ distance measuring domain shift and $\lambda$ is the error of an ideal joint hypothesis defined as $h^* = \arg\min_{h \in \mathcal{H}} \mathcal{E}_s(h) + \mathcal{E}_t(h)$, such that

$$\lambda = \mathcal{E}_s(h^*) + \mathcal{E}_t(h^*) \tag{4}$$

In DANN, training the discriminator on the DIRep bounds the $\mathcal{H}\Delta\mathcal{H}$ distance while training the feature extractor and the classifier on the source labeled data minimizes the error on the source domain ($\mathcal{E}_s(h)$)(see the proof in Ganin et al. (2016)). The third term, $\lambda$, is assumed to be sufficiently small in their analysis. However, as previous work has shown (Liu et al., 2019; Chen et al., 2019), the error of the ideal joint hypothesis $h^*$, especially for the target domain $\mathcal{E}_t(h^*)$, can not be overlooked in DANN. We present a reasonable explanation for this. In an unsupervised DA task, where the target data lacks labels, the classifier tends to take advantage of source-specific information that helps with source classification. Consequently, information that could be beneficial for classifying the target data may be omitted from the DIRep, leading to an increased $\mathcal{E}_t(h^*)$.

MaxDIRep addresses this pitfall by (1) decomposing the full representation into DIRep and DDRep, ensuring that together they encompass all necessary information to reconstruct the original data, and simultaneously (2) aligning the DDRep distribution with a standard normal distribution to minimize its information content. (3) extracting domain invariant features by using adversarial techniques. By doing so, the DIRep can capture more relevant domain-invariant features useful for target classification, as the information in DDRep is minimized. The improved target representation can lower the generalization error on the target domain, hence reducing $\mathcal{E}_t(h^*)$. As a result, the third term $\lambda$ is further bounded in our approach, yielding a lower bound for $\mathcal{E}_t(h)$ than DANN. We will justify this in Section 4.1.1 (See Figure 3b for the error rate of an ideal joint hypothesis trained using representations learned by DANN, DSN, and MaxDIRep).

## 3.3 Comparing MaxDIRep to DSN: MaxDIRep has a stronger constraint than DSN

Both DSN and MaxDIRep are based on decomposing the data representation into DIRep and DDRep. The main difference[1] is that instead of using $\mathcal{L}_{kl}$ to force the DDRep to contain minimal information as in MaxDIRep, DSN uses a linear orthogonality constraint between the private and shared representations of each domain. Formally, The constraint ($\mathcal{L}_{diff}$) is achieved by minimizing the dot products of DDRep ($DD^{S/T}$) and DIRep ($DI$) of source ($S$) and target ($T$) data respectively: $L_{diff} = \left\| DI \cdot DD^S \right\|^2 + \left\| DI \cdot DD^T \right\|^2$.

However, the orthogonality constraint does not always lead to a unique and optimal decomposition. For example, a different but also orthogonal or nearly orthogonal decomposition into DDRep and DIRep would be to minimize the domain-invariant information in DIRep with most image details contained in the DDRep. This decomposition, as discussed in Subsection 4.2, leads to poor DA performance but is not ruled out in the DSN algorithm due to its weaker linear orthogonality constraint.

---

[1]DSN also uses different neural networks to create the DDRep from their source and target.

To gain intuition about the difference between DSN and MaxDIRep, we looked at a 3-D geometrical analogy of a representation decomposition as shown in Figure 2. The source (S) and target (T) data represented in this analogy by vectors in 3D space are decomposed into the sum of DIRep ($DI$) and DDRep ($DD$): $S = DI_x + DD_x^S, T = DI_x + DD_x^T$ where the subscript $x$ represents the DSN (D) and MaxDIRep (V) algorithms, respectively. In DSN, the linear orthogonality constraint, $DI_D \cdot DD_D^{S,T} = 0$, enforces $DI_D \perp DD_D^{S,T}$, which can be satisfied by any points on the blue circle in Figure 2. In MaxDIRep, however, the size of DDRep's, i.e., $||S - DI|| + ||T - DI||$ is minimized, leading to a unique solution $DI_V$ (red dot in Figure 2), which not only satisfies the orthogonality constraint ($DI_V \perp DD_V^{S,T}$) but also maximizes the magnitude of DIRep ($||DI_V|| \geq ||DI_D||$) (see Appendix J for proof details).

This 3D geometric analogy suggests that the orthogonality constraint is weaker than minimizing the size of DDRep. Depending on the initialization, the system with only the orthogonality constraint can result in a sub-optimal solution (any point on the circle other than the MaxDIRep solution $DI_V$) that has poorer DA performance. For example, as shown in Figure 2, the origin, i.e., $DI_D = 0$, is a valid solution for DSN that satisfies the orthogonality constraint. Obviously, this extreme case solution with a minimal (zero) DIRep can not be used for DA at all.

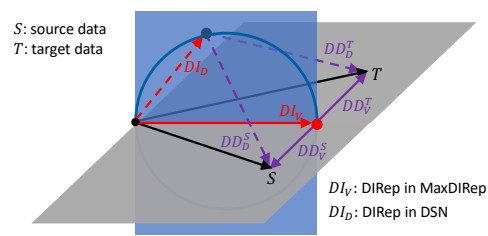

Figure 2: Schematic comparison between DSN and MaxDIRep. See text and Appendix J for explanation.

We expect the DA performance to become progressively worse as the DSN solution moves away from the solution obtained by MaxDIRep. Indeed, as we demonstrate later in Section 4.2 in a set of "mutual ablation" experiments in a realistic setting, if we perturb the DSN system by running DSN with a KL loss $\mathcal{L}_{kl}^{DI}$ applied to its DIRep for a certain time, DSN will find solutions that are consistent with the orthogonality constraint of DSN but have poorer DA performance. Furthermore, as we increase the strength of this perturbation, the DSN performance decreases, indicating the existence of many sub-optimal solutions for DSN, which is consistent with the geometric analogy (Figure 2). However, the opposite is not true, i.e., if we perturb the MaxDIRep system by applying a negative $\mathcal{L}_{diff}$ to make the DIRep and DDRep less orthogonal, MaxDIRep can still find the optimal solution with the same good DA performance.

## 4 Experiments

We evaluate MaxDIRep across different adaptation settings. In Subsection 4.1, we first construct synthetic datasets to explicitly demonstrate the advantage of MaxDIRep over DANN and DSN, which can use information specific to the source domain for classification and thus lead to poor DA performance. Specifically, we introduce "cheating information" that can be used easily for classification in the source domain but not in the target domain. This cheating information (or spurious correlation) could encourage a system to create a DIRep that is domain-invariant but does not have sufficient information about the target labels, leading to poor DA performance.

Next, in Subsection 4.2, we design a set of mutual ablation experiments between MaxDIRep and DSN to show that the key reason for the better performance of MaxDIRep than DSN is due to its stronger constraint of minimizing irrelevant domain-specific information than the orthogonality constraint of DSN.

In Subsection 4.3, we compare the performance of MaxDIRep on a set of standard benchmark datasets including MNIST (LeCun et al., 1998), MNIST-M (Ganin et al., 2016), Street View House Number (Netzer et al., 2011), synthetic digits (Ganin et al., 2016) and Office-31 (Saenko et al., 2010). We also assess MaxDIRep using the challenging Office-Home datasets (Venkateswara et al., 2017), which consist of four distinct domains, each containing 65 classes. Although the primary focus of this work is to compare our method with DANN and DSN, we also include comparisons with several recent methods on the Office-31 and Office-Home datasets to illustrate the practical value of our approach. Overall, our approach achieves better or similar results across standard DA benchmark datasets.

Finally, in Subsection 4.4, we demonstrate the application of MaxDIRep in training network intrusion detectors, building on the GAN-based algorithm by Singla et al. (2020), which successfully addressed the label scarcity issue in this domain using DA. Our findings show that MaxDIRep consistently improves the GAN-based results from Singla et al. (2020) and outperforms the performance of DSN and DANN, which highlights the versatility of MaxDIRep for non-image classification tasks.

## 4.1 Synthetic benchmarks and training methods

### 4.1.1 Synthetic benchmark based on Fashion-MNIST

Fashion-MNIST is a well-known dataset that we use as the source domain. We construct a target domain by flipping the original images by $180^o$. To add the cheating information, we add a one-hot vector to the source dataset that contains the correct classification (label). We call that information cheating bits. Specifically, each source image is reshaped into a $1 \times N$ vector, where $N$ represents the total number of pixels. The cheating bits (a one-hot vector of its label) are then appended to this image data vector. To the target dataset, we also add some bits to the flattened image data vector. The cheating bits in the target data have the same distribution as those in the source data, but they are not the labels of the target data. The idea is that if an algorithm were to use the cheating bits to classify the data, it would perform perfectly in the source data but poorly in the target data. We used two different ways of implementing the cheating bits in the target data: one is to use a random label (random cheating); and the other is to use the next label from the correct label (shift cheating).

**Benchmark algorithms** We compare our method against the following adversarial learning-based DA algorithms: GAN-based approach (Singla et al., 2020), Domain-Adversarial Neural networks (DANN) (Ganin et al., 2016) and Domain-Separation-Networks (DSN) (Bousmalis et al., 2016). We implemented both MaxDIRep and the explicit DDRep algorithm in the zero-shot setting. The explicit DDRep algorithm and the non-explicit DDRep achieve almost identical performance. We also provide two baselines: a classifier trained on the source domain samples without DA (which gives us the lower bound on target classification accuracy) and a classifier trained on the target domain samples (which gives us the upper bound on target classification accuracy). We compare the mean accuracy of our approach and the other DA algorithms on the target test set in Table 1. The z-scores of the comparison of our method with other methods are shown in Table 7 in the Appendix. More details of the topology, learning rate, hyper-parameters setup and results analysis are provided in Appendix B.

Table 1: Mean classification accuracy (%) of different adversarial learning-based DA approaches on the synthetic Fashion-MNIST benchmark.

| Model | No cheating | Shift cheating | Random cheating |
|---|---|---|---|
| Source-only | 20.0 | 11.7 | 13.8 |
| GAN-based (Singla et al., 2020) | 64.7 | 58.2 | 54.8 |
| DANN (Ganin et al., 2016) | 63.7 | 58.0 | 53.6 |
| DSN (Bousmalis et al., 2016) | 66.8 | 63.6 | 57.1 |
| MaxDIRep/Explicit DDRep | **66.9** | **66.8** | **61.6** |
| Target-only | 88.1 | 99.8 | 87.9 |

**The effect of single-bit DDRep** One particularly useful feature of the explicit DDRep algorithm is that it allows us to check the effect of the DDRep directly by flipping the domain bit ($d \to 1 - d$). This feature is highlighted in Figure 3a in the case of rotated Fashion-MNIST classification. The original images for the source and target domains are shown in columns 1 and 4, respectively. The reconstructed images are shown as columns 2 and 6 with the domain bit $d$ set to reflect their corresponding domains, i.e., $d = 0$ for column 2, $d = 1$ for column 6. Remarkably, by flipping the domain bit ($d \to 1 - d$) while keeping the DIRep unchanged, the resulting images (columns 3 and 5) resemble images from the other domain, which clearly demonstrates

the effectiveness of the minimal domain-specific information in DDRep (domain bit in the explicit DDRep model).

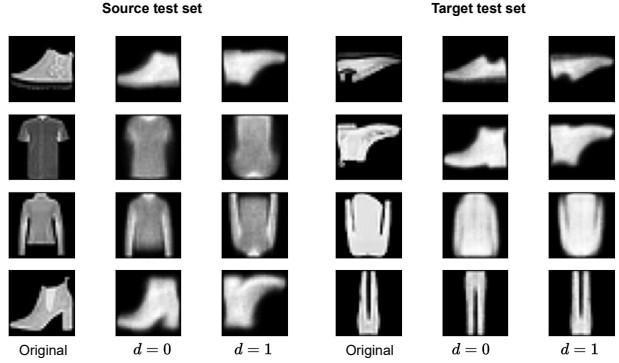

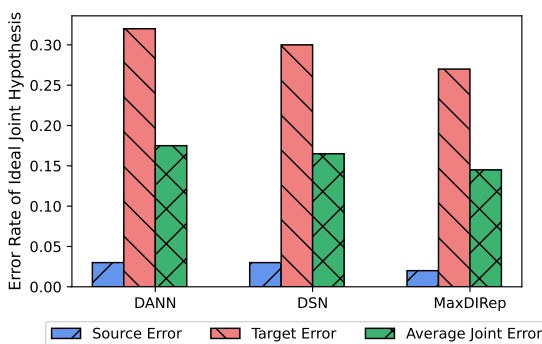

(a) Effects of flipping the domain bit. Columns 1 and 4 are the original images; columns 2 and 6 show reconstructions of originals; columns 3 and 5 show reconstructions with the domain bit flipped.

(b) The error rate of the ideal joint hypothesis trained using representations learned by DANN, DSN, and MaxDIRep.

Figure 3: Comparison of the reconstruction effect and the error of the ideal joint hypothesis.

**The error of an ideal joint hypothesis**  We follow the same approach in the literature to find the ideal joint hypothesis (Chen et al., 2019; Liu et al., 2019) on this dataset. Specifically, we train a new MLP classifier using the DIReps learned by DANN, DSN and MaxDIRep, respectively. The MLP classifier is trained on both source and target training data with labels, while each DA model is fixed. The target labels are only used for evaluating the error of the ideal joint hypothesis and are not involved in training the DA models. We then obtain the error rate of the trained MLP classifier on the source test set and target test set and calculate the average error rate. The results in Figure 3b show that MaxDIRep achieves the lowest error rate for the ideal joint hypothesis across both domains, thereby establishing a lower error bound for the target domain as indicated by **Theorem 1**.

### 4.1.2 Synthetic benchmark based on CIFAR-10

We are interested in more natural DA scenarios where the source and target images might be captured with different sensors and thus have different wavelengths and colors. To address this use case, we create another cheating benchmark based on CIFAR-10 with different color planes. We introduce the cheating color plane, where the choice of the color planes in the source data has a spurious correlation with the labels, while such correlation is absent in the target domain. Specifically, we create a source set with cheating color planes by encoding CIFAR-10 labels (0-9). For odd labels, only the blue channel is retained with probability ($p$), and either the blue or red channel is kept randomly for the rest. For even labels, only the red channel is retained with probability ($p$), and either the red or blue channel is kept randomly for the rest. The parameter ($p$) controls the spurious correlation strength between image color and label. In the target domain, only the green channel is retained for each CIFAR-10 image. We compare our approach with others using ($p$) values from $\{0, 0.2, 0.4, 0.6, 0.8, 0.9, 1.0\}$, where a larger ($p$) indicates a higher spurious correlation, making domain adaptation more challenging.

Table 2 presents the mean accuracy of MaxDIRep and the baseline algorithms on the target test set in a zero-shot setting. We used the full MaxDIRep model due to its better performance. The z-scores of the comparison of our method with other methods are shown in Table 11 in the Appendix. We observe similar performance degradation for the DANN, DSN and GAN-based approaches on this benchmark, suggesting that the adaptation difficulties of previous methods and the better results achieved by our method are not limited to a particular dataset. Due to space limits, the details of the experiments are given in Appendix D.

Table 2: Averaged classification accuracy (%) of different adversarial learning-based DA approaches on the synthetic CIFAR-10 dataset with a spectrum of bias.

| Model | 0% bias | 20% bias | 40% bias | 60% bias | 80% bias | 90% bias | 100% bias |
|---|---|---|---|---|---|---|---|
| Source-only | 10.0 | 10.0 | 10.0 | 10.0 | 10.0 | 10.0 | 10.0 |
| GAN-based (Singla et al., 2020) | 63.0 | 62.5 | 61.4 | 56.9 | 53.2 | 44.5 | 30.1 |
| DANN (Ganin et al., 2016) | 62.7 | 62.0 | 61.0 | 56.5 | 52.2 | 42.9 | 29.1 |
| DSN (Bousmalis et al., 2016) | 68.7 | 67.9 | 67.3 | 67.5 | 64.5 | 61.7 | 32.2 |
| MaxDIRep | **70.4** | **69.8** | **69.8** | **69.7** | **68.2** | **64.1** | **34.2** |
| Target-only | 78.9 | 78.9 | 78.9 | 78.9 | 78.9 | 78.9 | 78.9 |

As an additional experiment, we evaluate MaxDIRep and other approaches in a few-shot setting: the model is provided with a majority of unlabeled target data and a small amount of labeled target data. The results are shown in Figure 6 in the Appendix, and the training setup is described in Appendix D.3. We found that while the methods benefit from a small number of target labeled samples, MaxDIRep improves the most, surpassing DNS and GAN-based results by 12% and 25%, respectively, with only a total of 50 target labels.

## 4.2 The mutual ablation experiment between DSN and MaxDIRep

In DSN, the orthogonality constraint is enforced by a difference loss ($\mathcal{L}_{diff}$), while minimizing the information content of DDRep in MaxDIRep is enforced by a KL loss ($\mathcal{L}_{kl}$) for the DDRep. To demonstrate the difference between DSN and MaxDIRep, we designed mutual ablation experiments to answer the following questions:

- If we add a negative difference loss ($-\mathcal{L}_{diff}$) to MaxDIRep, would the performance of MaxDIRep decrease?

- On the other hand, if we add a KL loss for the DIRep ($\mathcal{L}_{kl}^{DI}$) in DSN, which acts as the opposite of the KL loss for the DDRep as in MaxDIRep, how would that affect the performance of DSN?

In the two sets of ablation experiments (shaded blue and yellow respectively in Table 3), we perturb the systems by adding the KL loss for DIRep ($\lambda_p \mathcal{L}_{kl}^{DI}$) and the inverse difference loss ($-\lambda_p \mathcal{L}_{diff}$) to DSN and MaxDIRep, respectively. Here, $\lambda_p$ represents the strength of the perturbation. We use one large and one small value of $\lambda_p = 0.001, 0.1$ (rows 2&4 for DSN, and rows 7&9 for MaxDIRep in Table 3) to explore the dependence on the perturbation strength. We then turn off these perturbations and continued the training until convergence to investigate if the systems can recover their original DA performance (rows 3&5 for DSN, and rows 8&10 for MaxDIRep in Table 3). For reference, we also list the performance when using the source data alone, DSN, and MaxDIRep in rows 1, 6, and 11, respectively in Table 3.

Table 3: Results of the ablation experiments conducted on the synthetic benchmark based on Fashion-MNIST. See the text for a detailed description.

| Methods | No cheating | Shift cheating | Random cheating |
|---|---|---|---|
| 1. Source only | 20.0 | 11.7 | 13.8 |
| 2. DSN + $\lambda_p \mathcal{L}_{kl}^{DI}$ ($\lambda_p = 0.001$) | 61.2 | 59.5 | 53.8 |
| 3. DSN* from 2 | 62.7 | 60.3 | 55.9 |
| 4. DSN + $\lambda_p \mathcal{L}_{kl}^{DI}$ ($\lambda_p = 0.1$) | 18.3 | 12.7 | 12.1 |
| 5. DSN* from 4 | 32.6 | 29.7 | 14.0 |
| 6. DSN | 66.8 | 63.6 | 57.1 |
| 7. MaxDIRep $-\lambda_p \mathcal{L}_{diff}$ ($\lambda_p = 0.001$) | **66.8** | **66.8** | 60.1 |
| 8. MaxDIRep* from 7 | **66.9** | **66.8** | 60.2 |
| 9. MaxDIRep $-\lambda_p \mathcal{L}_{diff}$ ($\lambda_p = 0.1$) | 63.6 | 63.6 | 60.1 |
| 10. MaxDIRep* from 9 | 65.5 | 65.5 | 60.3 |
| 11. MaxDIRep | **66.9** | **66.8** | **61.6** |

The findings in row 2 of Table 3 indicate that when we minimize the information content in DIRep during DSN training, DDRep and DIRep maintain orthogonality as evidenced by $\mathcal{L}_{diff} = 0$ in the experiment (see Table 8 in the Appendix). However, even this weak perturbation results in a worse DA performance than the original DSN. The results also show that even after this perturbation is removed (row 3), the optimal DA is not regained. This is consistent with the geometric analogy (Figure 2), which shows that many solutions satisfy the orthogonal constraint, but not all are equally good in DA. Here, DSN finds a sub-optimal solution from the initiation of weights reached by a weak "ablation" perturbation. Additionally, if we apply a stronger perturbation (row 4 in Table 3), the DSN algorithm becomes equivalent to a source-only DA scheme. Notably, the values for reconstruction loss and difference loss do not increase, and the classification loss on the source data is minimal (see the reported loss values in Table 8). This implies that DIRep predominantly carries the label information for the source and random information for the target, while DDRep retains the information necessary for reconstruction. Another important observation is that the KL losses on DIRep in the ablation experiments for DSN (rows 2&3) with the smaller perturbation strength ($\lambda_p = 0.001$) are significantly larger than those with the stronger perturbation ($\lambda_p = 0.1$, rows 4&5) (the loss values are reported in in Table 8). This confirms that a better DA is achieved with a higher information content in DIRep.

On the contrary, the performance of MaxDIRep is largely unaffected by the perturbation regardless of its strength (rows 7-10 in Table 3). This is because minimizing the information content of DDRep in MaxDIRep imposes a much stronger constraint, which contains the weaker orthogonal constraint imposed by $\mathcal{L}_{diff}$. This is additionally supported by the observation that $\mathcal{L}_{diff} = 0$ in the ablation experiments for MaxDIRep (see Table 9 in the Appendix).

### 4.3 Standard DA image benchmarks

There are two types of standard benchmark datasets: *type-1* datasets that present the same information in a different form, perhaps changing color or line width; *type-2* datasets that contain additional information in one domain, like the presence of the background of the object, which is absent in the other. It is clear that *type-2* datasets are prone to cheating while *type-* datasets are not. We apply MaxDIRep in three representative benchmark datasets: the digits dataset (*type-1*), the Office-31 dataset (*type-2*), and the Office-Home dataset (*type-2*). We find that MaxDIRep has a good performance comparable with other adversarial learning-based DA algorithms for the *type-1* dataset, while it outperforms other methods for the *type-2* dataset. We believe that outside of the setting of benchmarks, there are many more *type-2* datasets where MaxDIRep has a clear advantage.

**Digits datasets**    In this experiment, we use three DA pairs: 1) MNIST $\rightarrow$ MNIST-M, 2) Synth Digits $\rightarrow$ SVHN, and 3) SVHN $\rightarrow$ MNIST. Example images from all four datasets are provided in Appendix E. The architecture and hyper-parameter settings are also provided in Appendix E. Since the digits datasets are small datasets, we include the results in Table 12 in the Appendix, which shows the results on the digits datasets in the zero-shot setting. In summary, MaxDIRep outperforms all the other approaches we compare in all three DA scenarios.

**Office-31 dataset**    The most commonly used dataset for DA in object classification is Office-31 (Saenko et al., 2010). The Office dataset has 4110 images from 31 classes in three domains: Amazon (2817 images), Webcam (795 images) and DSLR (498 images). The three most challenging domain shifts reported in previous works are DSLR to Amazon ($D \rightarrow A$), Webcam to Amazon ($W \rightarrow A$) and Amazon to DSLR ($A \rightarrow D$). In $D \rightarrow A$ and $W \rightarrow A$ are the cases with the least labels in the source domain.

We follow previous work (Tzeng et al., 2017; Chen et al., 2020), which uses a pretrained ResNet-50 on ImageNet (Deng et al., 2009) as a base. We present the results for four zero-shot adaptation tasks in Table 4. We use the full MaxDIRep model due to its better performance. MaxDIRep is competitive on this adaptation task, matching the performance of Long et al. (2018) in $A \rightarrow D$ and $W \rightarrow D$, and outperforming all the approaches in all other tasks. However, it is worth noting that Long et al. (2018) utilizes a conditional discriminator conditioned on the cross-covariance of domain-specific feature representations and classifier predictions, which has the potential to improve our results further. We leave exploring this possibility for future work. Our approach shows the most significant performance improvements in scenarios such as $D \rightarrow A$

Table 4: Mean classification accuracy (%) of different baseline approaches on the Office-31 dataset. The results are cited from each study when available. The results of MCD (Saito et al., 2018) is cited from (Ma et al., 2021). We present our DSN replication results on the Office-31 dataset, which had not been evaluated by DSN.

| Model | $D \to A$ | $W \to A$ | $W \to D$ | $A \to D$ |
|---|---|---|---|---|
| Source-only | 62.5 | 60.7 | 98.6 | 68.9 |
| DANN (Ganin et al., 2016) | 68.2 | 67.4 | 99.2 | 79.7 |
| ADDA (Tzeng et al., 2017) | 69.5 | 68.9 | 99.6 | 77.8 |
| CDAN (Long et al., 2018) | 70.1 | 68.0 | **100.0** | **89.8** |
| GTA (Sankaranarayanan et al., 2018) | 72.8 | 71.4 | 99.9 | 87.7 |
| SimNet (Pinheiro, 2018) | 73.4 | 71.8 | 99.7 | 85.3 |
| MCD (Saito et al., 2018) | 71.0 | 67.2 | 98.4 | 84.1 |
| GPDA (Kim et al., 2019) | 72.3 | 68.8 | **100** | 85.5 |
| AFN (Xu et al., 2019) | 69.8 | 69.7 | 99.8 | 87.7 |
| Chadha et al. (Chadha & Andreopoulos, 2019) | 62.2 | - | - | 80.9 |
| IFDAN-1 (Deng et al., 2021) | 69.2 | 69.4 | 99.8 | 80.1 |
| DSN (Bousmalis et al., 2016) | 67.2 | 67.5 | 98.0 | 82.0 |
| MaxDIRep | **73.8** | **72.5** | **100.0** | 89.0 |

Table 5: Averaged accuracy (%) of different DA approaches on the Office-Home dataset. The results of DANN (Ganin et al., 2016) and CDAN (Long et al., 2018) are cited from (Long et al., 2018). The results of MCD (Saito et al., 2018) and GPDA (Kim et al., 2019) are cited from (Ma et al., 2021).

| Methods | Ar-Cl | Ar-Pr | Ar-Rw | Cl-Ar | Cl-Pr | Cl-Rw | Pr-Ar | Pr-Cl | Pr-Rw | Rw-Ar | Rw-Cl | Rw-Pr | Avg |
|---|---|---|---|---|---|---|---|---|---|---|---|---|---|
| Source-only | 34.9 | 50.0 | 58.0 | 37.4 | 41.9 | 46.2 | 38.5 | 31.2 | 60.4 | 53.9 | 41.2 | 59.9 | 46.1 |
| DANN (Ganin et al., 2016) | 45.6 | 59.3 | 70.1 | 47.0 | 58.5 | 60.9 | 46.1 | 43.7 | 68.5 | 63.2 | 51.8 | 76.8 | 57.6 |
| CDAN (Long et al., 2018) | 49.0 | 69.3 | 74.5 | 54.4 | 66.0 | 68.4 | 55.6 | 48.3 | 75.9 | 68.4 | 55.4 | 80.5 | 63.8 |
| MCD (Saito et al., 2018) | 45.6 | 60.9 | 69.2 | 50.8 | 60.7 | 60.5 | 46.2 | 44.0 | 74.7 | 62.6 | 53.8 | 77.5 | 58.6 |
| GPDA (Kim et al., 2019) | 47.1 | 62.0 | 70.4 | 53.6 | 62.3 | 60.9 | 49.7 | 47.2 | 72.3 | 63.7 | 54.0 | 78.6 | 60.2 |
| MaxDIRep | **53.5** | **71.1** | **78.9** | **54.9** | **66.0** | **68.8** | **59.5** | **48.7** | **78.6** | **69.5** | **56.6** | **80.8** | **65.6** |

and $W \to A$, in which background information is present within the $D$ and $W$ domains while being absent in the $A$ domain.

**Office-Home dataset**   Office-Home - a more difficult dataset than Office-31, consists of 15,500 images in 65 object classes, forming four extremely dissimilar domains (see Figure 8 in the Appendix for example images): Artistic images (Ar), Clip Art (Cl), Product images (Pr), and Real-World images (Rw). We use the same ResNet-50 network with the same training protocols and the hyperparameters from CDAN  (Long et al., 2015). More details can be found in Appendix G.

Strong results are also achieved on the Office-Home dataset as reported in Table 5 for the full MaxDIRep. In the evaluation of 12 transfer tasks, MaxDIRep consistently outperforms DANN (Ganin et al., 2016), CDAN (Long et al., 2018), MCD (Saito et al., 2018), and GPDA (Kim et al., 2019). We cannot find published results for ADDA and DSN on this benchmark. The classification accuracy of the Office-Home dataset is lower compared to the Office-31 dataset. The four domains in Office-Home have more categories and greater visual dissimilarity, making DA more difficult.

### 4.4   Application in Network Intrusion Detection (NID)

We also evaluate MaxDIRep in a non-image classification task, specifically in training network intrusion detectors. The NID datasets comprise network features extracted from both malicious and benign network traffic flows. A NID detector is then trained on these data to predict whether an incoming network flow is benign or originates from a network attack. However, most of the data is typically unlabeled and requires domain experts to analyze and label the traffic manually.

The GAN-based method proposed by Singla et al. (2020) addresses label scarcity in NID datasets via DA. It transfers knowledge from a labeled source NID dataset to a target NID dataset that contains a few labeled samples and many unlabeled data. These datasets can be created for different network types using various network protocols. An illustrative use case involves an organization that maintains a labeled source dataset comprising attack samples originating from its internal WiFi network. The organization then collects a limited set of attack samples to establish a target dataset specific to its Internet of Things (IoT) network. The organization applies DA techniques to utilize the source and target datasets for training an NID model for identifying attack traffic within the IoT network.

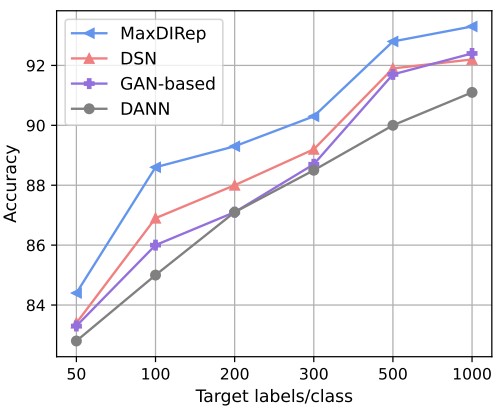

Figure 4: Mean classification accuracy on UNSW-NB15 test-set in few-shot setting.

Following their experimental design, we use NSL-KDD (Ahmed et al., 2009) as the source dataset and UNSW-NB15 (Moustafa & Slay, 2015) as the target dataset. The model is trained in a few-shot setting, utilizing all labeled source samples and a small amount of labeled target data. We replicate the experimental setup described in their paper and compare the performance of DANN, DSN, and MaxDIRep with implementation details provided in Appendix I. The results are reported in Figure 4, where we provide labels for 50, 100, 200, 300, 500 and 1000 target samples per class (benign and attack) during training. All methods improve with more target labels, maintaining the performance order: MaxDIRep > DSN > GAN-based > DANN.

## 5 Discussion, conclusion and future work

What is the intuitive reason for the better performance of MaxDIRep compared to previous methods, such as DSN, which shares the basic architecture? Neural networks are "lazy" as they tend to find the easiest solution (Chizat et al., 2020). Without the discriminator, the generator would be forced by the classifier to put the simplest information in DIRep to train the classifier for the source data, e.g., the snowy background in pictures of wolves or the "cheating" bit in our synthetic Fashion-MNIST dataset. Such a source-only classifier performs poorly in the target domain, as expected. A discriminator was introduced in previous methods, such as DANN, to solve this problem. However, as shown in this paper, having a discriminator is not enough. Specifically, the generator can evade the discriminator by generating random (spurious) information in the DIRep for the target data with the same distribution as the source data but does not correlate with the target label. An extreme case corresponds to the scenario where the DIRep contains only the correct labels for source data and random labels for target data, and the DDRep contains the rest of the information needed for reconstruction. This extreme case scenario leads to a poor solution, which is not prevented in the DSN algorithm due to its weak orthogonality constraint. On the contrary, our MaxDIRep algorithm, with its new loss function, minimizes information specific to each domain to rule out such poor solutions and creates a DIRep with sufficient information for good DA performance.

The general intuition described above is verified by using ablation experiments on a synthetic dataset and making a geometrical analogy. Indeed, by creating a maximal DIRep that contains genuine domain-independent information, MaxDIRep performs better than previous methods across all the standard benchmark datasets we tested. The hidden information effect is more likely to appear in complex datasets, e.g., we see more of its impact in CIFAR-10 than in Fashion-MNIST. The hidden information effect is also likely to appear when there is a drift in data, making classification more difficult. We adapt MaxDIRep and DSN for network intrusion detection using source and target datasets from different networks with significant data drift. MaxDIRep consistently outperforms both previous results and DSN.

An interesting future work direction is related to the use of pseudo-labeling, a powerful technique using pseudo-labels to provide noisy but sufficiently accurate labels for target data with which to progressively

update the model (Chen et al., 2020; Zou et al., 2018). Although the use of pseudo-labels is not considered in this work, it would be interesting to adopt this technique in our model. Since the initial estimate of the target label based on MaxDIRep is better than other algorithms, it is reasonable to expect that the more accurate initialization of pseudo-labeling, facilitated by our loss functions, should further improve the DA performance.

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

# Appendix

## Table of Contents

## A  More details on loss functions

Our code is available at (https://anonymous.4open.science/r/Maximal-Domain-Independent-Representations-Improve-Transfer-Learning-A422/README.md).

We provide the details of all the loss functions mentioned in Section 3 of the main paper. Recall that the data is given by $(x, l, d)$ where $x$ is the input, with $x^s$ and $x^t$ representing the source and target data, respectively. $l$ is the label of the sample, and $d$ is the domain identity.

In unsupervised DA, the classification loss applies only to the source domain, and it is defined as follows:

$$\mathcal{L}_c = -\sum_{i=1}^{N_s} l_i^s \cdot log \hat{l}_i^s \tag{5}$$

where $N_s$ represents the number of samples from the source domain, $l_i^s$ is the one-hot encoding of the label for the source input $x_i^s$, and $\hat{l}_i^s$ is the softmax output of $C(G(x_i^s))$.

The discriminator loss trains the discriminator to predict whether the DIRep is generated from the source or the target domain. $N_t$ represents the number of samples from target domain and $\hat{d}_i$ is the output of $D(G(x_i))$.

$$\mathcal{L}_d = - \sum_{i=1}^{N_s+N_t} \left\{ d_i log \hat{d}_i + (1 - d_i) log(1 - \hat{d}_i) \right\} \qquad (6)$$

The generator loss is the GAN loss with inverted domain truth labels:

$$\mathcal{L}_g = - \sum_{i=1}^{N_s+N_t} \left\{ (1 - d_i) log \hat{d}_i + d_i log(1 - \hat{d}_i) \right\} \qquad (7)$$

For the reconstruction loss, we use the standard mean squared error loss calculated from both domains:

$$\mathcal{L}_r = \sum_{i}^{N_s} ||x_i^s - \hat{x}_i^s||_2^2 + \sum_{i}^{N_t} ||x_i^t - \hat{x}_i^t||_2^2 \qquad (8)$$

where $\hat{x}_i^s = F(G(x_i^s), E(x_i^s))$ and $\hat{x}_i^t = F(G(x_i^t), E(x_i^t))$

Finally, the KL-divergence loss measures the distance between the distribution of DDRep, which we assume comes from a normal distribution with mean $\mathbb{E}(DDRep)$ and variance $\mathbb{V}(DDRep)$ and the standard normal distribution.

$$\mathcal{L}_{kl} = D_{KL}(Pr(DDRep) \parallel \mathcal{N}(0, I)) = -\frac{1}{2}(1 + log[\mathbb{V}(DDRep)] - \mathbb{V}(DDrep) - \mathbb{E}(DDRep)^2)$$

## B   Experiment details on Fashion-MNIST

### B.1   Network architecture

All the methods are trained using the Adam optimizer with the learning rate of $2e - 4$ for $10,000$ iterations. We use batches of 128 samples from each domain for a total of 256 samples. When training with our model (MaxDIRep), the label prediction pipeline (generator and classifier) has eight fully connected layers (FC1, ..., FC7, FC_OUT). The number of neurons in FC1-4 is 100 for each layer. FC5 is a 100-unit layer that generates DIRep, followed by two 400-unit layers (FC6-7). FC_OUT is the output layer for label prediction. The discriminator and decoder each have four layers with 400 hidden units, followed by the domain prediction layer and reconstruction layer, respectively. The encoder has two layers with 400 units, followed by 100-unit $z\_mean$, 100-unit $z\_variance$, and sampling layer. Each of the 400-unit layers uses a ReLU activation function.

All the other models have the same architecture as MaxDIRep when applicable. For the GAN-based approach and DANN, we turn off the decoder and corresponding losses. For the DSN, we keep the same network architecture for common networks and use $\mathcal{L}_g$ for the similarity loss. Furthermore, we implement the shared and private encoders with the same shape output vectors (Bousmalis et al., 2016).

### B.2   Hyperparameters

As suggested in previous work (Ganin et al., 2016), the coefficient of the loss, which encourages domain invariant representation, should be initialized as 0 and changed to 1. We use the following schedule for the coefficient of $\mathcal{L}_g$ in all the experiments where $t$ is the training iteration:

$$\lambda = \frac{2}{1 + exp(-t)} - 1 \qquad (9)$$

The increasing coefficient allows the discriminator to be less sensitive to noisy signals at the early stages of the training procedure. For other hyperparameters, we used $\beta = 0.1, \gamma = 0.15, \mu = 0.1$ (the hyperparameters were not tuned using validation samples).

We closely follow the setup of weights of the loss functions used in the DSN paper (Bousmalis et al., 2016) and DANN paper (Ganin et al., 2016). To boost the performance of DSN, we set the coefficient of $\mathcal{L}_{recon}$ to 0.15 and the coefficient of $\mathcal{L}_{diff}$ to 0.05, tuned parameter values determined by Bousmalis et al. (2016) using a validation set of target labels. To make a fair comparison, we use the same schedule for the coefficient of $\mathcal{L}_g$ and set the coefficient of $\mathcal{L}_c$ to 0.1 in DSN.

### B.3 Results and analysis

Table 6 summarizes the mean classification accuracy of different approaches for three cheating scenarios. In the no cheating scenario, we use the original Fashion-MNIST as the source and flip the Fashion-MNIST for the target. We report the z-score of the comparison of the mean classification accuracy of our method with the mean classification accuracy of other methods over five independent runs (see Table 7). The higher the z-score, the more statistical confidence we should have that our method outperforms the other methods. A z-score of 2.33 corresponds to 99% confidence that our method is superior.

Table 6: Mean classification accuracy (%) of different adversarial learning-based DA approaches on the constructed Fashion-MNIST dataset.

| Model | No cheating | Shift cheating | Random cheating |
|---|---|---|---|
| Source-only | 20.0 | 11.7 | 13.8 |
| GAN-based (Singla et al., 2020) | 64.7 | 58.2 | 54.8 |
| DANN (Ganin et al., 2016) | 63.7 | 58.0 | 53.6 |
| DSN (Bousmalis et al., 2016) | 66.8 | 63.6 | 57.1 |
| MaxDIRep | **66.9** | **66.8** | **61.6** |
| Target-only | 88.1 | 99.8 | 87.9 |

Table 7: Z-test score value comparing MaxDIRep to other models on the constructed Fashion-MNIST dataset. z>2.3 means the probability of MaxDIRep being no better is ≤0.01.

| Model | No cheating | Shift cheating | Random cheating |
|---|---|---|---|
| GAN-based (Singla et al., 2020) | 1.55 | 3.28 | 3.68 |
| DANN (Ganin et al., 2016) | 2.26 | 4.17 | 4.33 |
| DSN (Bousmalis et al., 2016) | 0.16 | 2.60 | 3.18 |

In the no cheating scenario, MaxDIRep outperforms GAN-based and DANN and matches the result of DSN. The performance of GAN-based and DANN results in a 5% accuracy drop for the shift cheating and a 10% drop for the random cheating. This validates our concern: the source cheating bits can be picked up in the DIRep as they represent an easy solution for the classifier trained only with source samples. If so, the cheating generator will perform poorly for the target domain, which has different cheating bits. Our method has only 0.1% and 5% accuracy drop respectively. As a reconstruction-based method, DSN performs better in the presence of cheating bits. In the shift and random cheating, our approach significantly outperforms DSN with a z-score of 2.60 and 3.18, respectively, which shows the correctness of our intuition that minimizing the information content of DDRep can result in transferring as much information as possible to the DIRep. In the explicit DDRep algorithm, the DDRep is minimal as it only contains the domain label. Given a richer DIRep, our method improves DA performance on the target data.

## C Loss values in the mutual ablation study

We provide the details of the loss values in the mutual ablation experiment in Section 4.2. Table 8 shows the effect of KL loss for DIRep $\lambda_p \mathcal{L}_{kl}^{DI}$ on DSN's loss functions. Table 9 shows the effect of the inverse difference loss $-\lambda_p \mathcal{L}_{diff}$ to MaxDIRep's loss functions. We made the following observations:

- From both tables, we do not observe any significant increase in other loss values compared to the regular DSN (line 6 in Table 8) and MaxDIRep (line 11 in Table 9).

- When we reduce the DIRep during DSN training, $\mathcal{L}_{diff}$ is always 0, which implies that DDRep and DIRep maintain orthogonality.

- $\mathcal{L}_{diff}$ loss is always zero in Table 9. This implies the orthogonality between DIRep and DDRep in MaxDIRep.

- In lines 2 and 3, the KL losses on DIRep are significantly larger than in lines 4 and 5. If we look at Table 3 in the main text, 2 and 3 also achieve much better DA, which shows that a DIRep with more information improves DA performance.

Table 8: Effect of KL loss for DIRep $\lambda_p \mathcal{L}_{kl}^{DI}$ to DSN's loss functions. The loss values reported here are the average data from both the source and the target.

| Methods | No cheating | | | Shift cheating | | | Random cheating | | |
|---|---|---|---|---|---|---|---|---|---|
| | $\mathcal{L}_{kl}^{DI}$ | $\mathcal{L}_{recon}$ | $\mathcal{L}_{diff}$ | $\mathcal{L}_{kl}^{DI}$ | $\mathcal{L}_{recon}$ | $\mathcal{L}_{diff}$ | $\mathcal{L}_{kl}^{DI}$ | $\mathcal{L}_{recon}$ | $\mathcal{L}_{diff}$ |
| 2. DSN + $\lambda_p \mathcal{L}_{kl}^{DI}$ ($\lambda_p = 0.001$) | 29.7 | 0.04 | 0 | 19.7 | 0.04 | 0 | 25.8 | 0.05 | 0 |
| 3. DSN* from 2 | 41.5 | 0.04 | 0 | 48.6 | 0.04 | 0 | 30.6 | 0.05 | 0 |
| 4. DSN + $\lambda_p \mathcal{L}_{kl}^{DI}$ ($\lambda_p = 0.1$) | 1.725 | 0.05 | 0 | 1.65 | 0.05 | 0 | 2.04 | 0.06 | 0 |
| 5. DSN* from 4 | 16 | 0.05 | 0 | 14.3 | 0.04 | 0 | 11.9 | 0.06 | 0 |
| 6. DSN | N/A | 0.04 | 0 | N/A | 0.04 | 0 | N/A | 0.05 | 0 |

Table 9: Effect of the inverse difference loss $-\lambda_p \mathcal{L}_{diff}$ to MaxDIRep's loss functions. The loss values reported here are the average data from both the source and the target.

| Methods | No cheating | | | Shift cheating | | | Random cheating | | |
|---|---|---|---|---|---|---|---|---|---|
| | $\mathcal{L}_{kl}$ | $\mathcal{L}_{recon}$ | $\mathcal{L}_{diff}$ | $\mathcal{L}_{kl}$ | $\mathcal{L}_{recon}$ | $\mathcal{L}_{diff}$ | $\mathcal{L}_{kl}$ | $\mathcal{L}_{recon}$ | $\mathcal{L}_{diff}$ |
| 7. MaxDIRep $-\lambda_p \mathcal{L}_{diff}$ ($\lambda_p = 0.001$) | 0 | 0.07 | 0 | 0 | 0.07 | 0 | 0 | 0.07 | 0 |
| 8. MaxDIRep* from 7 | 0 | 0.07 | 0 | 0 | 0.07 | 0 | 0 | 0.07 | 0 |
| 9. MaxDIRep $-\lambda_p \mathcal{L}_{diff}$ ($\lambda_p = 0.1$) | 0 | 0.07 | 0 | 0 | 0.07 | 0 | 0 | 0.07 | 0 |
| 10. MaxDIRep* from 9 | 0 | 0.07 | 0 | 0 | 0.07 | 0 | 0 | 0.07 | 0 |
| 11. MaxDIRep | 0 | 0.07 | N/A | 0 | 0.07 | N/A | 0 | 0.07 | N/A |

## D   Experiment details on CIFAR-10

The source set with cheating color planes is constructed as follows. First, we encode labels in CIFAR-10 with values between 0 and 9. Then, for each CIFAR-10 image, if its label is odd, we keep only the B channel with prob $p$ and randomly keep the B or the R channel for the rest. Similarly, if the label is even, with prob $p$, the image has only the R color channel, and either the R or B channel is kept for the rest. For example, when $p = 1$, all images with odd labels have only the B channel, and all images with even labels have only the R channel. We call $p$ the *bias* since it controls the strength of the spurious correlation between the image's color and its label. In the target domain, for each CIFAR-10 image, we keep only the G channel regardless of the label. We compare our approach and the others with $p$, taking values from the set $\{0, 0.2, 0.4, 0.6, 0.8, 0.9, 1.0\}$. A larger value of $p$ indicates a higher level of spurious correlation in the source data and, thus, a more challenging DA task.

In this "cheating-color-plane" setting, the GAN-like algorithms might cheat by leveraging the correlation between the presence or absence of the color planes and the image's label to create an easier classification scheme for the labeled source data. Consequently, the DIRep would include false cheating clues, which can degrade performance on the target data where the cheating clues lead to the wrong answer.

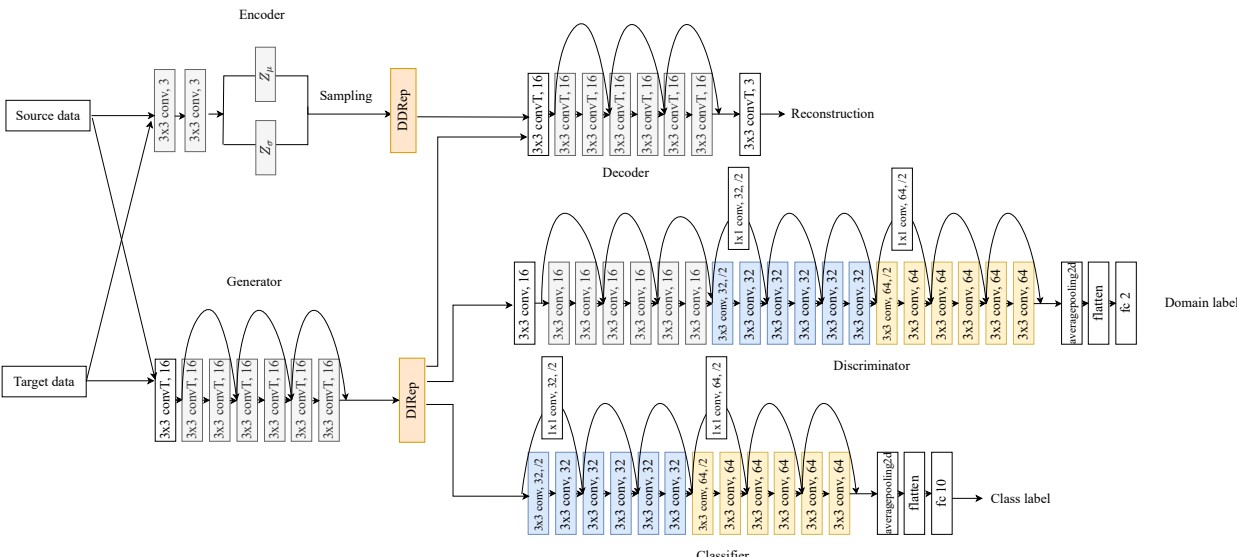

Figure 5: CIFAR-10 training architecture; inspired by the classical ResNet-20 (He et al., 2016)

### D.1 Network architecture and training procedure

When training with our approach, we implement the network components as deep residual neural networks (ResNets) with short-cut connections (He et al., 2016). ResNets are easier to optimize and sometimes gain accuracy from increased depth. For our approach, we implemented the full MaxDIRep. The architecture is shown in Figure 5. The label prediction pipeline is adopted from the ResNet 20 for CIFAR-10 in He et al. (2016). For the generator, the first layer is $3 \times 3$ convolutions. Then, we use a stack of 6 layers with $3 \times 3$ convolutions on the feature maps of size 32. The number of filters is 16. The architecture of the classifier consists of a stack of $6 \times 2$ layers with $3 \times 3$ convolutions on the feature maps of sizes $\{16, 8\}$ respectively. To maintain the network complexity, the number of filters is $\{32, 64\}$. The classifier ends with a global average pooling and a fully connected layer with softmax.

For the discriminator, the network inputs are $32 \times 32 \times 16$ domain invariant features. The first layer is $3 \times 3$ convolutions. Then we use a stack of $6 \times 3$ layers with $3 \times 3$ convolutions on the feature maps of sizes 32, 16, and 8, respectively, with 6 layers for each feature map size. The numbers of filters is $\{16, 32, 64\}$ respectively. The discriminator ends with a global average pooling, a 2-way fully connected layer, and softmax.

The encoder has 4 convolutional layers: three $3 \times 3$ filters, two $3 \times 3$ filters, two $3 \times 3$ filters ($z$ mean) and two $3 \times 3$ filters ($z$ variance) respectively. A sampling layer is also implemented, which outputs the DDRep from the latent distribution $z$. The decoder learns to reconstruct an input image using its DIRep and DDRep together. The configuration of the decoder is the inverse of that of the generator.

We implemented the same ResNet-based architecture for all other approaches (when applicable). We use a weight decay of 0.0001 and adopt the BN (Ioffe & Szegedy, 2015) for all the experiments. We use the same schedule in Section B.2 for the coefficient of $\mathcal{L}_g$ in all the experiments. For other hyperparameters, we used $\beta = 1, \gamma = 1, \mu = 1/2000$ in MaxDIrep and set the coefficient of $\mathcal{L}_{recon}$ to 0.15, the coefficient of $\mathcal{L}_{diff}$ to 0.05, and the coefficient of $\mathcal{L}_c$ to 1 in DSN.

### D.2 Results and analysis

We report the mean accuracy of different DA methods and our approach on the target test set in Table 10. The z-scores of comparing our method with other methods are shown in Table 11.

For all the DA tasks with varying biases, we observe that our approach outperforms the other approaches in terms of accuracy in the target test set. This improvement is most pronounced when the source set has 60%

Table 10: Averaged classification accuracy (%) of different adversarial learning-based DA approaches on the constructed CIFAR-10 dataset with a spectrum of bias.

| Model | 0% bias | 20% bias | 40% bias | 60% bias | 80% bias | 90% bias | 100% bias |
|---|---|---|---|---|---|---|---|
| Source-only | 10.0 | 10.0 | 10.0 | 10.0 | 10.0 | 10.0 | 10.0 |
| GAN-based (Singla et al., 2020) | 63.0 | 62.5 | 61.4 | 56.9 | 53.2 | 44.5 | 30.1 |
| DANN (Ganin et al., 2016) | 62.7 | 62.0 | 61.0 | 56.5 | 52.2 | 42.9 | 29.1 |
| DSN (Bousmalis et al., 2016) | 68.7 | 67.9 | 67.3 | 67.5 | 64.5 | 61.7 | 32.2 |
| MaxDIRep | **70.4** | **69.8** | **69.8** | **69.7** | **68.2** | **64.1** | **34.2** |
| Target-only | 78.9 | 78.9 | 78.9 | 78.9 | 78.9 | 78.9 | 78.9 |

Table 11: z-test score value comparing MaxDIRep to other models on the constructed CIFAR-10 dataset. z>2.3 means the probability of MaxDIRep being no better than the other models is ≤0.01.

| Model | 0% bias | 20% bias | 40% bias | 60% bias | 80% bias | 90% bias | 100% bias |
|---|---|---|---|---|---|---|---|
| GAN-based (Singla et al., 2020) | 5.23 | 3.20 | 5.93 | 12.8 | 11.31 | 7.20 | 4.58 |
| DANN (Ganin et al., 2016) | 5.44 | 3.42 | 6.22 | 13.2 | 12.02 | 7.79 | 5.70 |
| DSN (Bousmalis et al., 2016) | 2.68 | 3.00 | 3.95 | 3.47 | 7.43 | 3.78 | 2.23 |

and 80% bias levels, which means that over half of the source data has a spurious correlation between their color planes and labels. The poor performance of the GAN-based and DANN approaches is another example of how the generator in these approaches learns a DIRep that depends on the spurious correlation. This false representation leads to an effect similar to over-fitting, where the model performs well on the source data but does not generalize well on the target data where the same correlation does not exist. In the DSN approach, the shared representation contains some domain-independent information other than the cheating clues which helps classification in the target domain.

### D.3  Few-shot DA

As an additional experiment, we also evaluated the proposed algorithm in a few-shot DA setting on the constructed CIFAR-10 dataset. The model is provided with a majority of unlabeled target data and a small amount of labeled target data. In our setting, we revealed $1, 5, 10, 20, 50$ and $100$ labels per class, which we then used to contribute to the classification loss through the label prediction pipeline. We also provided the same number of labels for the GAN-based and DSN methods. We skipped the DANN method since its performance is very similar to that of the GAN-based approach. More importantly, we ask the following question: *How much does each algorithm gain from a small labeled target training set for different biases?* The classification loss on the target ensures that the generator does not get away with learning a DIRep that contains only the cheating clue, which could cause a high classification loss on the target.

We select the four most representative biases and show the results in Figure 6. For 40%, 60% and 80% biases, the classification accuracy does improve, but not significantly, as the number of target labels increases. The performance order of MaxDIRep > DSN > GAN-based is preserved. When the bias is equal to 100%, the performance curves are quite different. All of them increase significantly with the number of target labels while the order of performance is preserved. While all three algorithms benefit from a small number of target labeled samples, MaxDIRep improves the most, surpassing DNS and GAN-based results by 12% and 25% respectively, with only a total of 50 target labels (note that it corresponds to 5 labels/class in Figure 6).

## E   SVHN, MNIST, MNIST-M and Synth Digits

We evaluate the empirical performance of MaxDIRep on four widely used DA benchmarks: MNIST (LeCun et al., 1998), MNIST-M (Ganin et al., 2016), Street View House Number (Netzer et al., 2011) and synthetic

---

[2]We present the results from our replication of DSN using regular MSE loss, which match the values reported in the DSN paper. However, our attempts to replicate their results using scale-invariant MSE were unsuccessful. Other attempts (fungtion, 2024) at replication were less successful than ours. Nonetheless, comparing results using the same reconstruction loss provides the most accurate and fair comparison.

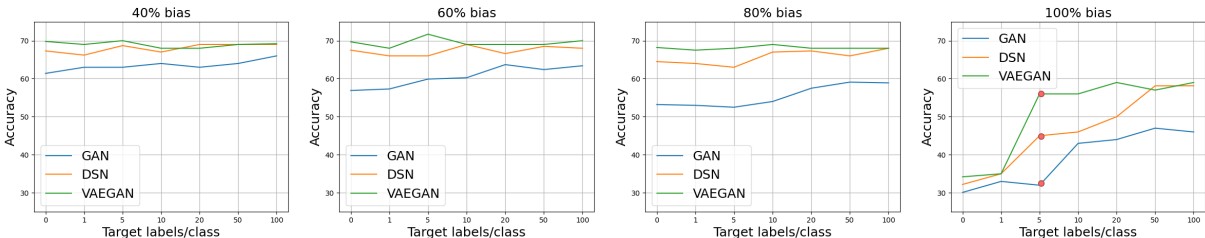

Figure 6: Mean classification accuracy on CIFAR-10 with the few-shot setting for three different DA algorithms. Overall, a few target labels improve classification accuracy. The improvement is significant in a 100% bias setting.

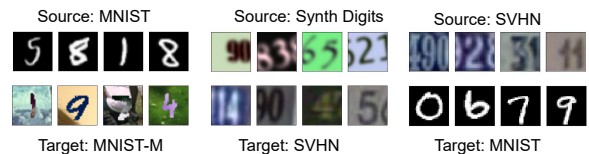

Figure 7: Example images from four DA benchmark datasets for three scenarios.

digits (Ganin et al., 2016). We use three DA pairs: 1) MNIST → MNIST-M, 2) Synth Digits → SVHN, and 3) SVHN → MNIST. Example images from all four datasets are provided in Figure 7. We implemented our CNN topology based on the ones used in (Bousmalis et al., 2016) and (Ganin et al., 2016). We used Adam with the learning rate of 0.0002 for 25,000 iterations. The batch size is 128 for each domain. We did not use validation samples to tune hyperparameters. To make fair comparisons, we follow the instructions in Bousmalis et al. (2016) and activate the $\mathcal{L}_g$ after 20,000 training steps. For other hyperparameters, we used $\beta = 1$, $\gamma = 1$, and $\mu = 1$.

**MNIST to MNIST-M.** We use the MNIST dataset as the source domain and a variation of MNIST called MNIST-M as the target. MNIST-M was created by blending digits from the original MNIST set over patches randomly extracted from color photos from BSDS500 (Arbelaez et al., 2010).

**Synthetic Digits to SVHN.** This scenario is widely used to demonstrate the algorithm's effectiveness when training on synthetic data and testing on real data. We use synthetic digits as the source and Street-View House Number (SVHN) as the target.

**SVHN to MNIST.** In this experiment, we further increase the gap between the two domains. The digit shapes in SVHN are quite distinct from those handwritten digits in MNIST. Furthermore, SVHN contains significant image noise, such as multiple digits in one image and a blurry background.

Table 12 shows the results on the digits datasets in the zero-shot setting. We skipped the explicit DDRep because the full MaxDIRep model performs better. In summary, MaxDIRep outperforms all the other approaches we compared in all three DA scenarios.

## F  Office-31 dataset

Office-31 dataset comprises three distinct domains: Amazon, DSLR, and Webcam. Example images from all four datasets are provided in Figure 8. We used the ResNet-50 architecture pretrained on the ImageNet dataset as the generator, following a common approach in recent DA studies (Tzeng et al., 2017; Chen et al., 2020). This choice allowed us to leverage the knowledge gained from ImageNet's large-scale dataset and apply it to our DA task. We used Adam with the learning rate of 0.0002. The batch size is 16 for each domain. We did not use validation samples to tune hyperparameters and set them to be the same values as in previous

Table 12: Mean classification accuracy (%) of different adversarial learning-based DA approaches on the digits datasets. The results are cited from each study when available.

| Methods | MNIST to MNIST-M | Synth Digits to SVHN | SVHN to MNIST |
|---|---|---|---|
| Source-only | 56.6 | 86.7 | 59.2 |
| DANN (Ganin et al., 2016) | 76.6 | 91.0 | 73.8 |
| ADDA (Tzeng et al., 2017) | 80.0 | - | 76.0 |
| DSN$^2$ (Bousmalis et al., 2016) | 80.4 | 89.0 | 79.5 |
| MaxDIRep | **82.0** | **91.2** | **85.8** |

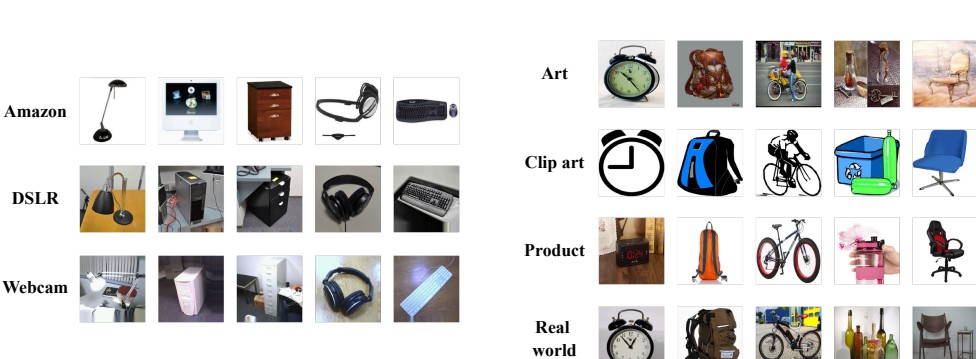

Figure 8: Example images from different domains in Office-31 and Office-Home.

works (Bousmalis et al., 2016; Ganin et al., 2016) when available. We used $\lambda = 0.1$, $\beta = 1$, $\gamma = 0.05$, and $\mu = 1/2000$.

## G  Office-Home dataset

The office-Home dataset comprises four extremely dissimilar domains: Artistic images, Clip Art, Product images, and Real-World images. Example images from all four datasets are provided in Figure 8. We follow the standard protocols for unsupervised DA (Long et al., 2018; Bousmalis et al., 2016). Similar to the setup in Office-31, we opted to utilize the ResNet-50 architecture pretrained on the ImageNet dataset as the generator. We used Adam with the learning rate of 0.0002. The batch size is 16 for each domain. We set the hyperparameters to be the same as the ones used in (Long et al., 2018; Bousmalis et al., 2016): we activate the $\mathcal{L}_g$ after 25 epochs of training and set $\lambda = 0.1$ after 25 epochs. We use $\beta = 1$, $\gamma = 0.05$, $\mu = 1/2000$.

## H  KL divergence analysis

We reported the KL divergence from some of our experiments in Table 13.

## I  Network intrusion detection dataset

For this evaluation, we use NSL-KDD as the source dataset and UNSW-NB15 as the target dataset. We remove the specific categories of attacks from the datasets and model this as a binary classification problem, i.e., predicting whether the current record belongs to the attack or benign category. Since the source and target datasets have different numbers of features, we apply PCA to both datasets to transform them into datasets with 100 features each before training.

Table 13: We report the KL divergence ($\mathcal{L}_{kl}$) from our experiments, calculated as the average over data from both the source and target domains.

| Task | KL divergence ($\mathcal{L}_{kl}$) | Task | KL divergence ($\mathcal{L}_{kl}$) |
|---|---|---|---|
| Fashion-MNIST (no cheating) | 9.53e-07 | Office-31 (W → A) | 0.03 |
| Fashion-MNIST (shift cheating) | 1.25e-06 | Office-31 (W → D) | 0.03 |
| Fashion-MNIST (random cheating) | 1.13e-06 | Office-31 (A → D) | 0.05 |
| CIFAR-10 (40% bias) | 4.80e-06 | Office-Home (Ar → Cl) | 0.13 |
| CIFAR-10 (60% bias) | 3.20e-06 | Office-Home (Ar → Rw) | 0.10 |
| Office-31 (D → A) | 0.07 | Office-Home (Rw → Cl) | 1 |

We use the same network topologies for MaxDIRep and all other approaches mentioned in Appendix B.1. All the methods are trained using the Adam optimizer with the learning rate of $2e-4$ for $10,000$ iterations. We use batches of 128 samples from each domain for a total of 256 samples. We directly used the reported result from Singla et al. (2020) for the GAN-based method.

To avoid noises during the early stages of training, we activate the $\mathcal{L}_g$ after 5000 epochs of training and set $\lambda = 0.1$ for the remaining training steps. For other hyper-parameters, we used $\beta = 1$, $\gamma = 0.1$ and $\mu = 1/2000$ (the hyperparameters were not tuned using validation samples).

To make a fair comparison with DSN, we set the coefficient of $\mathcal{L}_{recon}$ to 0.1 and the coefficient of $\mathcal{L}_{diff}$ to 0.001. We use the same schedule for the coefficient of $\mathcal{L}_g$ and set the coefficient of $\mathcal{L}_c$ to 1 in DSN. For DANN, We use the same schedule for the $\mathcal{L}_g$ and set the coefficient of $\mathcal{L}_c$ to 1.

## J   The geometrical interpretation of MaxDIRep versus DSN

To understand the difference between DSN and MaxDIRep, we looked at a 3-D geometrical interpretation of representation decomposition as shown in Figure 2 in the main text. Here, we show that all points on the blue circle satisfy the orthogonal condition, i.e., $DI_D \perp DD_D^{S,T}$.

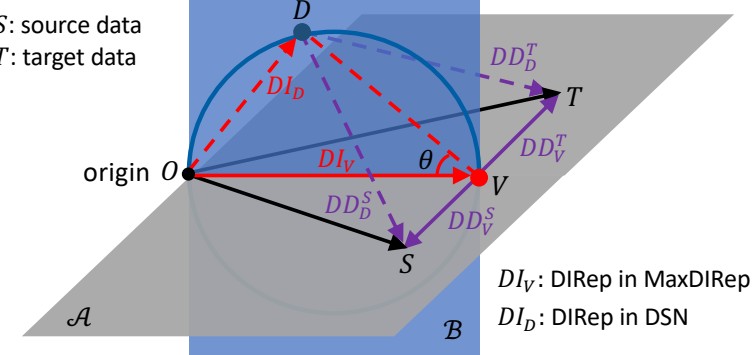

Figure 9: Schematic comparison between DSN and MaxDIRep.

The source and target data are represented by two vectors $S = \overrightarrow{OS}$, $T = \overrightarrow{OT}$ where $O$ is the origin, as shown in Figure 9. We assume the source and target vectors have equal amplitude $|\overrightarrow{OS}| = |\overrightarrow{OT}|$. Let us define the plane that passes through the triangle $O - S - T$ as plane-$\mathcal{A}$ (the gray plane in Figure 9). The mid-point between $S$ and $T$ is denoted as $V$. Let us draw another plane (the blue plane-$\mathcal{B}$) that passes through the line $OV$ and is perpendicular to the plane-$\mathcal{A}$. The blue circle is on the blue plane-$\mathcal{B}$ with a diameter given by $OV$. Denote an arbitrary point on the blue circle as D with the angle $\angle DVO = \theta$. Let us define the plane that passes through the triangle $D - S - T$ as plane-$\mathcal{C}$ (not shown in Figure 9).

Since the blue plane-$\mathcal{B}$ is the middle plane separating S and T, we have $ST \perp OV$ and $ST \perp DV$ (note that $XY$ represents the line between the two points $X$ and $Y$). Therefore, the line $ST$ is perpendicular to the whole plane-$\mathcal{B}$: $ST \perp \mathcal{B}$, which means that $ST$ is perpendicular to any line on plane-$\mathcal{B}$. Since the line $DV$ is on the plane-$\mathcal{B}$, we have $OD \perp ST$. Since $OV$ is the diameter of the blue circle, we have $OD \perp DV$. Since $DV$ and $ST$ span the plane-$\mathcal{C}$, we have $OD$ is perpendicular to the whole plane-$\mathcal{C}$: $OD \perp \mathcal{C}$, which means that $OD$ is perpendicular (orthogonal) to any line on plane-$\mathcal{C}$ including $DS$ and $DT$. Therefore, we have proved: $OD \perp DS$, $OD \perp DT$.

Note that with the notation given here, we can express the DIRep and DDRep for MaxDIRep (V) and DSN (D) as

$$DI_V = \overrightarrow{OV}, \;\; DD_V^S = \overrightarrow{VS}, \;\; DD_V^T = \overrightarrow{VT}.$$
$$DI_D = \overrightarrow{OD}, \;\; DD_D^S = \overrightarrow{DS}, \;\; DD_D^T = \overrightarrow{DT}.$$

Since we have proved that $OD \perp DS$, $OD \perp DT$ for any point $D$ on the blue circle, this means that any point on the blue circle satisfies the orthogonality constraint $DI_D \perp DD_D^{S,T}$.

In MaxDIRep, the size of DDRep's, i.e.,

$$||S - DI|| + ||T - DI|| = (||\overrightarrow{VS}||^2 + ||\overrightarrow{DV}||^2)^{1/2} + (||\overrightarrow{VT}||^2 + ||\overrightarrow{DV}||^2)^{1/2}$$

is minimized leading to an unique solution $DI_V$ shown as the red dot (point $V$) in Figure 9, which satisfies the orthogonality constraint ($DI_V \perp DD_V^{S,T}$) as it is on the blue circle. More importantly, the MaxDIRep solution is unique as it maximizes the magnitude of DIRep ($||DI_V|| \geq ||DI_D||$). This can be seen easily as follows. Given the angle $\angle DVO = \theta$, we have $||DI_D|| = ||DI_V|| \sin \theta \leq ||DI_V||$.

