# OpenReview forum: "Maximizing Information in Domain-Invariant Representation Improves Transfer Learning"
_TMLR — Rejected by TMLR_

### Review · Reviewer_LhYy · 2024-10-18

**Summary Of Contributions:**

This paper discusses a new domain adaptation (DA) algorithm designed to improve classification in target domains with limited labels. Traditional DA approaches, like Domain-Separation Networks (DSN), divide data into domain-independent (DIRep) and domain-dependent representations (DDRep), but DSN's weak constraint on orthogonality between these two can lead to poor performance, as useful information may remain in the DDRep. To solve this, the new algorithm imposes a stronger constraint using a KL divergence loss to minimize the DDRep, enhancing the DIRep for better transfer learning. The authors demonstrate their algorithm's robustness and superior performance on benchmark datasets and show its compatibility with pretrained models and network intrusion detection.

**Audience:**

Yes

**Broader Impact Concerns:**

There appear to be no major ethical concerns regarding the broader impact of this work. The authors have not explicitly addressed potential biases introduced during domain adaptation. It may be useful to include a section discussing the broader ethical implications, such as fairness and bias mitigation, especially if this method is applied to sensitive areas like healthcare or criminal justice.

**Claims And Evidence:**

Yes

**Requested Changes:**

Experiments: I strongly recommend some more powerful baselines comparing to the method proposed in this paper (see Strengths And Weaknesses) .

Theory: It will be better to include some discussion, but it's optional to explicitly provide a bound.

Typos (Notice the double quotation marks in your paper!)
- “transferred" in the fifth paragraph
- “ablation" in Section 4.2

**Strengths And Weaknesses:**

Strengths
- It's an interesting approach to minimize domain-dependent representations and the method get robust performance on standard benchmarks such as Fashion-MNIST, CIFAR-10, and Office datasets.
- The authors provide thorough experimental evaluations, including ablation studies and comparisons with conventional methods like Domain-Separation Networks (DSN). It seems like an improvement beyond DSN and the authors illustrate the idea with geometrical comparison. It's easy to understand and follow.

Weaknesses
- Can the method in this paper scales to larger dataset?
- The baseline algorithms are conventional but not strong enough. Maybe the authors can consider some other methods for comparison, such as [1-2]. The authors are not required to achieve the state-of-the-art performance. However, compared with some more advanced algorithms, even the algorithms of a few years ago can also enhance the application value of this algorithm. Otherwise, the algorithm in this article looks like an improvement on some classic algorithms from 8 years ago.
- I encourage the authors to provide some insights from the perspective of domain adaptation theory. For instance, information-theoretic bound [3].

[1] Saito K, Watanabe K, Ushiku Y, et al. Maximum classifier discrepancy for unsupervised domain adaptation[C]//Proceedings of the IEEE conference on computer vision and pattern recognition. 2018: 3723-3732.

[2] Kim M, Sahu P, Gholami B, et al. Unsupervised visual domain adaptation: A deep max-margin gaussian process approach[C]//Proceedings of the IEEE/CVF Conference on Computer Vision and Pattern Recognition. 2019: 4380-4390.

[3] Park G Y, Lee S W. Information-theoretic regularization for multi-source domain adaptation[C]//Proceedings of the IEEE/CVF international conference on computer vision. 2021: 9214-9223.

---

> ### Author Response · Authors · 2024-11-26
>
> We thank the reviewer for their insightful comments, as well as for kindly acknowledging our contributions. We are more than happy to address these comments and hope that the following response helps address some concerns. We have merged similar comments in weakness and requested changes.
>
> >Can the method in this paper scale to a larger dataset?
>
> While we agree with the reviewer that the application of our approach to larger image datasets such as medical images would be very interesting, we wish to retain our experiments of this paper on standard DA benchmarks (F-MNIST, CIFAR10, Digits, Office-31 and Office-Home) and leave the experimentation with large real-life datasets as future work. That said, we have included a novel application of our approach in network intrusion detection where the datasets we have tested are large: NSL-KDD has 125,973 record samples, and UNSW-NB15 has 175,341 data samples respectively.
>
>
>
> >The baseline algorithms are conventional but not strong enough. Maybe the authors can consider some other methods for comparison, such as [1-2]. The authors are not required to achieve state-of-the-art performance. However, compared with some more advanced algorithms, even the algorithms of a few years ago can also enhance the application value of this algorithm. Otherwise, the algorithm in this article looks like an improvement on some classic algorithms from 8 years ago.
>
>  As the reviewer kindly acknowledges, we proposed a method to overcome the tendency to use information specific to the source domain and discard relevant information for target classification when using DANN and DSN. However, we agree with the reviewer that comparing with more recent methods would better position our algorithm in the literature. Hence, we have included comparisons with several recent methods (including those two mentioned by the reviewer) on the Office-31 and Office-Home databases.   Overall, performance of our approach for all the benchmark datasets we studied compares favorably with the other algorithms including the new ones mentioned by the referee.
>
> **Changes made to the revised paper**
>
> - We have added two baseline algorithms mentioned by the reviewer in the evaluation of the Office-31 dataset, together with six other baseline algorithms from 2018-2021 (See Table 4 in the revised paper).
> -  We have added two baseline algorithms mentioned by the reviewer in the evaluation of the Office-Home dataset (See Table 5 in the revised paper).
>
>
> > I encourage the authors to provide some insights from the perspective of domain adaptation theory. For instance, information-theoretic bound [3].
>
> We would like to thank the referee for the suggestion. While providing the explicit bound turns out to be challenging, we have followed the referee's suggestion and analyzed our approach based on the formal domain adaptation theory from [1]. Specifically, we found that our approach conforms well with the DA theory. Furthermore, our analysis also provided some additional insights into why MaxDIRep yields better adaptability than DANN.  These theoretical insights have been substantiated through new experiments conducted similarly to those in [2] and [3]. We have revised our manuscript significantly to include this new analysis and the corresponding new experiments. Please refer to the revised manuscript for specific changes listed below.
>
> **Changes made to the revised paper**
>
> - We added a theoretical understanding of MaxDIRep based on the domain adaptation theory in Section 3.2.
> - We added a new experiment to justify our theoretical insights in Figure 3(b) with a detailed description in Section 4.1.1 under the paragraph "the error of an ideal joint hypothesis''.
>
>
>
> >Typos (Notice the double quotation marks in your paper!)
>
> We would sincerely thank the reviewer for their meticulous review. These are indeed typos, and we have fixed them throughout the paper in the updated PDF.
>
>
> Reference:
>
> [1] Shai Ben-David, John Blitzer, Koby Crammer, Alex Kulesza, Fernando Pereira, and Jennifer Wortman Vaughan. A theory of learning from different domains. Machine learning, 79:151–175, 2010.
>
> [2] Hong Liu, Mingsheng Long, Jianmin Wang, and Michael Jordan. Transferable adversarial training: A general approach to adapting deep classifiers. In International Conference on Machine Learning, pp. 4013–4022. PMLR, 2019.
>
> [3] Xinyang Chen, Sinan Wang, Mingsheng Long, and Jianmin Wang. Transferability vs. discriminability: Batch spectral penalization for adversarial domain adaptation. In International conference on machine learning, pp. 1081–1090. PMLR, 2019.

---

> > ### Comment · Reviewer_LhYy · 2024-11-28
> > **Thanks**
> >
> > Thank you for your detailed response with additional experiments and theoretical analysis, my major concern has been addressed.

---

### Review · Reviewer_R63S · 2024-10-19

**Summary Of Contributions:**

The paper introduces a novel algorithm called MaxDIRep for domain adaptation (DA), which aims to improve the transfer of learned classifiers between different domains. The core contribution is the decomposition of data representations into Domain Independent Representation (DIRep) and Domain Dependent Representation (DDRep). Unlike previous methods like Domain Separation Networks (DSN), MaxDIRep imposes a stronger constraint on the DDRep by using a KL divergence loss to minimize it, ensuring that the DIRep contains as much information as possible for classification. The paper demonstrates superior performance over state-of-the-art DA algorithms across several image benchmark datasets and a network intrusion detection task, showcasing the robustness of MaxDIRep.

**Audience:**

Yes

**Broader Impact Concerns:**

No related concerns.

**Claims And Evidence:**

Yes

**Requested Changes:**

1. The authors should address the contradiction regarding the role of domain-dependent information in target classification. Since DDRep should not contribute to classification across domains, the authors should either clarify their interpretation or revise the claim to avoid implying that information in domain-dependent representation contributes to generalization.
2. The idea of "maximizing" or "minimizing" representations (DIRep and DDRep) needs to be reformulated. Instead of maximizing DIRep or minimizing DDRep, the authors could discuss ensuring that DIRep captures relevant, domain-invariant features while minimizing irrelevant domain-specific information in DDRep.
3. The claim that applying KL divergence between $𝐸(𝑋)$ and a normal distribution forces input information into DIRep requires additional justification. The authors should provide a stronger explanation of how KL divergence influences the latent space.

**Strengths And Weaknesses:**

Strengths:
1. The method demonstrates better or comparable results against existing DA algorithms like DSN, DANN, and ADDA on benchmark datasets.
2. The algorithm performs well not only in standard image-based DA tasks but also in non-image domains, like network intrusion detection, highlighting its broader applicability.
3. The authors conduct mutual ablation experiments to show that their method's improvements are directly tied to the stronger constraint on DDRep.

Weaknesses:
1. The authors' claim that information useful for classification in target domains can exist in domain-dependent representations (DDRep) is contradictory, since domain-dependent information can not generalize across domains. Leveraging domain-dependent information for classifications across domains usually brings bias.
2. The idea of minimizing the Domain-dependent representations but maximizing domain invariant representations is not appropriate, since representation can not be maximized or minimized.
3. Applying KL divergence between $E(X)$ and normal distribution can not force input information into DIRep.
4. The norm of $DI_V$ larger than $DI_D$ proved in the paper does not convincingly demonstrate that it can make DDRep include less information and DIRep with more information, and further improve the performance of MaxDIRep with a larger norm of DIRep.
5. The statement that "a close to zero KL loss indicates that most of the weights are near zero" is incorrect. A small KL divergence does not necessarily mean that the weights in the model are near zero, nor does it imply that "the Domain-Dependent Representation (DDRep) is orthogonal or near orthogonal to the Domain-Invariant Representation (DIRep)".
6. Important reference missing: [1] Domain Adaptation with Invariant Representation Learning: What Transformations to Learn?

---

> ### Author Response · Authors · 2024-11-26
>
> We thank the reviewer for acknowledging our contributions and providing insightful comments on the presentation.  Below, we address these comments and clarify some of the key technical details (we have merged similar comments in weakness and requested changes).
>
> > The authors should address the contradiction regarding the role of domain-dependent information in target classification. Since DDRep should not contribute to classification across domains, the authors should either clarify their interpretation or revise the claim to avoid implying that information in domain-dependent representation contributes to generalization.
>
>
> We are not sure what the referee is referring to regarding "the contradiction regarding the role of domain-dependent information in target classification". In our approach, the DDRep is not used for classification; the classifier only uses the DIRep.  In a less-than-ideal separation of DIRep and DDRep, as in DSN, some information that could be useful in classifying the target domain can end up in the DDRep and hence can not be used in classifying the target data.  However, in our approach, DIRep retains sufficient information about the target labels because the DDRep is constrained by the KL divergence to have minimal information content.
>
>
> **Changes made to the revised paper**
> We have made it clear in the revised version the role of domain-dependent information in cross-domain generalization. Please see the introduction in the revised paper.
>
>
>
> >The idea of "maximizing'' or "minimizing'' representations (DIRep and DDRep) needs to be reformulated. Instead of maximizing DIRep or minimizing DDRep, the authors could discuss ensuring that DIRep captures relevant, domain-invariant features while minimizing irrelevant domain-specific information in DDRep.
>
> We apologize for the confusion caused by our terminology regarding maximizing and minimizing representations and thank the reviewer for bringing this to our attention. Our intention is to minimize information content in DDRep while ensuring that DIRep captures domain-invariant features useful for target classification. We have re-worked the wording and the title of the paper to make it clear that what we minimize is the information content in the DDRep.
>
> **Changes made to the revised paper**
> We have fixed our terminology throughout the paper. In addition, we have changed our title.
>
> >The claim that applying KL divergence between $E(X)$ and a normal distribution forces input information into DIRep requires additional justification. The authors should provide a stronger explanation of how KL divergence influences the latent space.
>
>
>  Our goal to achieve good adaptation is to develop a constraint on the DIRep extraction, preventing it from discarding information about the target labels. To enforce such a constraint, we use the data-generating process and the assumption of minimal domain-specific information across domains. This assumption is reasonable as Stojanov et al. [1] have the same assumption in their paper.  MaxDIRep prevents the discarding of valuable information about target labels by enforcing the minimal information content in the DDRep during the data generation process from both DDRep and DIRep.  We measure the KL divergence on DDRep with a standard normal distribution (which is considered as the baseline distribution that has little information) to constrain the information content in DDRep.  Since DDRep corresponds to minimal information specific to each domain, this forces DIRep to retain information about the target labels and, in turn, result in a better invariant representation.   We added a discussion on how the improved DIRep can further constrain the target error bound and provided an additional experiment to justify that the DIRep learned by MaxDIRep contains more useful information about the target labels than the DIReps learned by DSN and DANN.
>
>
>
> **Changes made to the revised paper**
> - We have added an explanation of how KL divergence influences the latent space in Section 3.
> - We added a theoretical understanding of MaxDIRep based on the domain adaptation theory [2] in Section 3.2.
> - We added a new experiment to justify our theoretical insights in Figure 3(b) with a detailed description in Section 4.1.1 under the paragraph "the error of an ideal joint hypothesis''.  The new experiment was designed following a similar approach as those in [3] and [4].

---

> ### Author Response · Authors · 2024-11-26
>
> >The norm of $DI_{V}$ larger than $DI_D$ proved in the paper does not convincingly demonstrate that it can make DDRep include less information and DIRep with more information and further improve the performance of MaxDIRep with a larger norm of DIRep.
>
> The purpose of the 3-D geometric analogy in which we can show the magnitude of $DI_V$ is larger than $DI_D$ is meant to demonstrate the difference between MaxDIRep and DSN and the fact that minimizing the information content in DDRep represents a stronger constraint than orthogonality. We agree that this by itself does not demonstrate that MaxDIRep performs better than DSN. However, motivated by the 3-D geometric analogy, we carried out the mutual ablation experiments (section 4.2), which did show that having a stronger constraint in MaxDIRep leads to a better performance than DSN. Specifically, our experiments revealed cases where the DIRep learned by DSN is domain-invariant but lacks predictive information for target labels. For instance, in Table 3, we showed that depending on initialization, DSN can find "bad" solutions with a DIRep that fails to represent the data for classification, as evidenced by target classification accuracy close to random, whereas MaxDIRep has a much more robust good performance. Notably, despite this failure in predictive performance, the reconstruction loss and difference loss remained low (see Table 8 in the Appendix for the corresponding loss values) in DSN. This demonstrates that label-related information is predominantly captured by the DDRep rather than the DIRep in DSN.
>
>
> >The statement that "a close to zero KL loss indicates that most of the weights are near zero'' is incorrect. A small KL divergence does not necessarily mean that the weights in the model are near zero, nor does it imply that ``the Domain-Dependent Representation (DDRep) is orthogonal or near orthogonal to the Domain-Invariant Representation (DIRep)''.
>
> We sincerely thank the reviewer for catching this mistake: we have removed the incorrect statement "a close to zero KL loss indicates that most of the weights are near zero''. As to the statement, "This means that MaxDIRep always results in a DDRep that is orthogonal or near orthogonal to its DIRep, and thus satisfies the orthogonality constraint of DSN'', it was made in the context of the 3-D geometric analogy (section 3.3), which is true, i.e., minimizing the size of DDRep naturally leads to a solution that satisfies the orthogonality constraint as a necessary condition (see Appendix J for the detailed proof). Consistent with this intuition gained from the 3-D geometric analogy, the results from our mutual ablation experiments show that $\mathit{L_{diff}= \left\Vert DI \cdot DD^{S}\right\Vert^2 + \left\Vert DI \cdot DD^{T}\right\Vert^2}$ is consistently small (near zero) in MaxDIRep (see Table 9 in the Appendix), which confirms that  MaxDIRep effectively enforces the orthogonality constraint.
>
> >Important reference missing: [1] Domain Adaptation with Invariant Representation Learning: What Transformations to Learn?
>
> We would like to thank the reviewer for bringing this work to our attention. We have cited this paper and discussed this paper in the related work section of the revised paper.
>
>
>
> References
>
> [1] Stojanov, Petar, Zijian Li, Mingming Gong, Ruichu Cai, Jaime Carbonell, and Kun Zhang. "Domain adaptation with invariant representation learning: What transformations to learn?." Advances in Neural Information Processing Systems 34 (2021).
>
>
> [2] Shai Ben-David, John Blitzer, Koby Crammer, Alex Kulesza, Fernando Pereira, and Jennifer Wortman Vaughan. A theory of learning from different domains. Machine learning, 79:151–175, 2010.
>
> [3] Hong Liu, Mingsheng Long, Jianmin Wang, and Michael Jordan. Transferable adversarial training: A general approach to adapting deep classifiers. In International Conference on Machine Learning, pp. 4013–4022. PMLR, 2019.
>
> [4] Xinyang Chen, Sinan Wang, Mingsheng Long, and Jianmin Wang. Transferability vs. discriminability: Batch spectral penalization for adversarial domain adaptation. In International conference on machine learning, pp. 1081–1090. PMLR, 2019.

---

### Review · Reviewer_kweN · 2024-11-05

**Summary Of Contributions:**

The paper proposes a new algorithm, MaxDIRep, to address the domain adaptation problem. Specifically, it decomposes data representation into domain-independent (DIRep) and domain-dependent (DDRep) components. Unlike previous methods, MaxDIRep imposes a stronger constraint to make DDRep resemble random noise, aiming to learn a DIRep that captures more information relevant to labels or classification tasks, thereby enhancing adaptation performance. Extensive ablation studies confirm its improvement over the baseline method, DSN. Additionally, the authors empirically demonstrate MaxDIRep’s effectiveness in discarding spurious correlations in domain adaptation. They also show that MaxDIRep outperforms existing methods on multiple image classification benchmark datasets, as well as a non-image classification task.

**Audience:**

Yes

**Claims And Evidence:**

Yes

**Requested Changes:**

Please refer to the weaknesses.

**Strengths And Weaknesses:**

### Strengths

- The experimental section is relatively comprehensive.

- The experimental results support the effectiveness of the proposed method.

### Weaknesses

- Limited Comparison with Recent Related Work: Representation decomposition or disentanglement has been widely studied in domain adaptation research. However, the paper lacks comparisons with recent related work and methods. In the related work section, the authors state, “These and other related topics fall outside the scope of our paper, and thus will not be addressed any further.” Why?

- Writing Lacks Academic Quality: The writing requires significant improvement in clarity and academic rigor. For example:

  - Abstract and Vague Descriptions: Many descriptions are abstract and vague. For instance, in the title, does “maximal Domain Independent Representation” mean that the DIRep has the maximal magnitude or contains the most information? This definition is unclear.

  - Lack of Conciseness: The writing is verbose, with some overly complex sentences, and is hard to understand. For instance, the last sentence of the third-to-last paragraph in the Introduction (“However,..., in their DDRep rather than in the DIRep”) is difficult to understand.

  - Insufficient Citations in Introduction: Some content in the Introduction lacks citations. For example, in the third paragraph of the Introduction, the phrase “Our general intuition, largely consistent with previous work” lacks clarity on which prior works are being referenced

- Lack of Experimental or Theoretical Support: Some descriptions lack experimental or theoretical support. For instance, the first sentence in Section 3.1, “We have used the KL divergence to measure the information content of the DDRep and found that the DDRep contains the equivalent of one bit of information or even less in some cases,” requires experimental results to support this claim. Similarly, in Section 4, the statement, “If MaxDIRep outperforms these foundational techniques, it is reasonable to anticipate that MaxDIRep will further improve the performance of other methods derived from these primary approaches,” needs justification.

- Insufficient Details on Experimental Setup: Important experimental details are insufficiently described. For example, in Section 4.1.1, “To add the cheating information, we add to the source dataset a one-hot vector that contains the correct classification (label),” it is unclear how exactly this was implemented.

---

> ### Author Response · Authors · 2024-11-26
>
> We would like to thank the reviewer for the comments. We hope that the following response helps highlight our contributions and clarify some of the technical details.
>
> We begin by addressing the concerns regarding comparison with more recent methods.
> > Limited Comparison with Recent Related Work: Representation decomposition or disentanglement has been widely studied in domain adaptation research. However, the paper lacks comparisons with recent related work and methods. In the related work section, the authors state, “These and other related topics fall outside the scope of our paper and thus will not be addressed any further.” Why?
>
> We would kindly bring to the reviewer's attention that one of the core components of this paper is to rigorously and systematically compare our approach to DANN and DSN, which are the two methods most closely related to ours. In particular, we show that DANN and DSN can hide relevant information for the target labels in the DDRep. We see more of the effect of this shortcoming when the source domain data has spurious correlations. Our method (MaxDIRep) avoids this problem by suppressing the information content in the DDRep, thus creating the DIRep with maximal information. To the best of our knowledge, the comparison of different methods on this issue has not been explored in the existing literature.  In the related work section,  we briefly discussed other methods that are less similar to our approach, e.g., the ones based on representation disentanglement, which focus on different problem settings (instead of one source domain and one target domain) or utilizes non-adversarial training techniques to learn domain-invariant features.  Since these other approaches are not as closely related to our method as DANN and DSN, we only mentioned them in the related work section without an in-depth comparison.
>
> However, we agree with the reviewer that comparing our paper with more recent methods would strengthen our paper. To address the reviewer's comment, in the revised version, we have included several recent approaches as additional baselines on the Office-31 and Office-Home datasets to strengthen the evaluation of our approach. Overall, we found that our method achieves comparable or outperforms these recent baseline algorithms.
>
> **Changes made to the revised paper**
> - We have added another eight baseline algorithms from 2018-2021 in the evaluation of the Office-31 dataset (see Table 4 in the revised paper).
> - We have added another two baseline algorithms in the evaluation of the Office-Home dataset (see Table 5 in the revised paper).
>
>
>
> > Abstract and Vague Descriptions: Many descriptions are abstract and vague. For instance, in the title, does “maximal Domain Independent Representation” mean that the DIRep has the maximal magnitude or contains the most information? This definition is unclear.
>
>  It means that DIRep contains more domain-invariant information useful for target classification. We have changed the title to "Maximizing Information in Domain-Invariant Representation Improves Transfer Learning''. We have gone through the abstract and other parts of the manuscript to make the descriptions clearer.
>
> >Lack of Conciseness: The writing is verbose, with some overly complex sentences, and is hard to understand. For instance, the last sentence of the third-to-last paragraph in the Introduction (“However,..., in their DDRep rather than in the DIRep”) is difficult to understand.
>
> We have changed this sentence in the revised version. This sentence conveys the point that the constraints in DSN do not guarantee that the learned DIRep has sufficient information for predicting labels in the target domain. We have gone through our manuscript to make it clearer.
>
>
> >Insufficient Citations in Introduction: Some content in the Introduction lacks citations. For example, in the third paragraph of the Introduction, the phrase “Our general intuition, largely consistent with previous work” lacks clarity on which prior works are being referenced.
>
> We have added citations ([1][2]) to support and clarify this statement.

---

> ### Author Response · Authors · 2024-11-26
>
> > Lack of Experimental or Theoretical Support: Some descriptions lack experimental or theoretical support. For instance, the first sentence in Section 3.1, “We have used the KL divergence to measure the information content of the DDRep and found that the DDRep contains the equivalent of one bit of information or even less in some cases,” requires experimental results to support this claim. Similarly, in Section 4, the statement, “If MaxDIRep outperforms these foundational techniques, it is reasonable to anticipate that MaxDIRep will further improve the performance of other methods derived from these primary approaches,” needs justification.
>
> We first clarify the comment regarding the first sentence in Section 3.1. Using the standard normal distribution as the reference, we use KL divergence to measure the relative information content in DDRep.  From the results of the full MaxDIRep algorithm, we found that the DDRep contains a small amount of information as measured by the KL divergence, which is consistently small in all the experiments. We have added this clarification at the beginning of Section 3.1 and included a new Table with the exact KL divergence of DDRep in some of our experiments in the Appendix (Please see Table 13).
>
> For the second statement, we intended to convey that combining our approach with pseudo-labeling would outperform the combination of pseudo-labeling with DANN and DSN since it is reasonable to assume that the estimated target labels based on our approach would be more accurate. However, we understand that this point may have been too speculative without experimental support. To avoid any confusion, we have removed this statement in the revised version of the paper.
>
>
> **Changes made to the revised paper**
>
> - We have added the KL divergence of DDRep from our experiments to support our claim (see Table 13 in the revised paper).
> - We have removed the second statement in the revised version of the paper.
>
>
>
>
> >Insufficient Details on Experimental Setup: Important experimental details are insufficiently described. For example, in Section 4.1.1, “To add the cheating information, we add to the source dataset a one-hot vector that contains the correct classification (label),” it is unclear how exactly this was implemented.
>
> Here, we provide details on how the datasets are constructed and have revised our paper accordingly.  To add the cheating information, we add a one-hot vector to the source dataset that contains the correct classification (label). We call that information cheating bits. Specifically, each source image is reshaped into a $1 \times N$ vector, where $N$ represents the total number of pixels. The cheating bits (a one-hot vector of its label) are then appended to this image data vector. To the target dataset, we also add some bits to the flattened image data vector. The cheating bits in the target data have the same distribution as those in the source data, but they are not the labels of the target data. The idea is that if an algorithm were to use the cheating bits to classify the data, it would perform perfectly in the source data but poorly in the target data. We used two different ways of implementing the cheating bits in the target data:  one is to use a random label (random cheating), and the other is to use the next label from the correct label (shift cheating).
>
> **Changes made to the revised paper**
>
> - We have added the detailed descriptions of the experimental setup such as the implementation of the cheating bits in the revised manuscript.
>
>
> Reference:
>
> [1] Stojanov, Petar, Zijian Li, Mingming Gong, Ruichu Cai, Jaime Carbonell, and Kun Zhang. "Domain adaptation with invariant representation learning: What transformations to learn?." Advances in Neural Information Processing Systems 34 (2021).
>
> [2] Bousmalis, Konstantinos, George Trigeorgis, Nathan Silberman, Dilip Krishnan, and Dumitru Erhan. "Domain separation networks." Advances in neural information processing systems 29 (2016).

---

> > ### Comment · Reviewer_kweN · 2024-12-21
> >
> > Thank you to the authors for their detailed responses, the improvements made in the manuscript, and the additional experiments. Many of my concerns have been addressed.
> >
> > However, I still have two points that require further consideration:
> >
> > - Presentation: While the revised manuscript is easier to read and understand, there are still instances of informal or non-academic language. For example, the last sentence in the first paragraph of the Related Work section: "Here, we briefly describe previous methods focusing on those that are closely related to ours." can be further shortened as "We summarize prior methods relevant to our approach."
> >
> > - Methodology and main contribution: The authors claim that maximizing information in DIRep benefits transfer learning, a point also reflected in the revised title. However, the definition of "information" in this context needs to be further clarified to enhance the paper's accuracy and rigor.

---

> > > ### Author Response · Authors · 2024-12-22
> > >
> > > Dear Reviewer kweN,
> > >
> > > Thank you for reviewing our revised manuscript and highlighting the sentence that could be improved. We will modify that sentence in the final version and thoroughly review the manuscript again to improve the presentation.
> > >
> > > Our claim is that maximizing the information content in domain-invariant representation (DIRep) improves transfer learning, as the DIRep retains sufficient information about the target labels. We measure the size of the information content in domain-dependent representation (DDRep) using the KL divergence from a standard normal distribution. We will modify the relevant parts of the manuscript to make the description clearer in the final version.
> > >
> > > We hope our responses have addressed your comments, though we are happy to address any further concerns. Thank you once again for your time and effort.

---

### Author Response · Authors · 2024-11-26
**Revised Manuscript**

Dear Action Editor and Reviewers,

We would like to thank you for the reviews and for providing the requested changes that helped us improve our paper. Along with our responses to each individual reviewer, we have included the revised version and a diff version that highlights the changes from the original (provided in the supplementary material) for your consideration.

---

### Decision · Action_Editor_9tSm · 2024-12-22

**Recommendation:** Reject

**Comment:**

This paper considered maximizing the information in the unsupervised domain adaptation. They further argued that  the information useful for classification in the target domain can “hide” in the domain dependent representation learning.

After rounds of discussions, all reviewers **unanimously** recommended the rejection or weak rejection due to the supporting evidence and below I would like to discuss the rationale.

- The paper writing lacks scientific rigor. **I agree.**  For example, (1) some reviewer pointed out that the idea of minimizing the Domain-dependent representations but maximizing domain invariant representations is not appropriate, since representation can not be maximized or minimized. From AE’s view, Indeed, a formal description in this context should be learning a representation that maximizes the correlation (or relation) between useful information and minimizes the correlation (or relation) between the nuisance information. (2) In the abstract, author claimed that ‘’However, information useful for classification in the target domain can “hide” in the DDRep.’’ I also have difficulty in understanding what ‘’hide’’ means.  I mean there are notable imprecise descriptions within the paper such that several reviewers feel quite confused about the rationale or motivation withins the paper. I think these significantly influence the perception of the actual paper quality and lead to a unanimous rejection.

- The paper may be beneficial from better experimental comparisons. I am **leaning to agree** with this proposition. Please take note that the title is designed for general domain adaptation rather than a specific Domain-Separation Networks (DSN). The additional experimental results would be definitely helpful for stronger supporting evidence.

- Some justifications. For example, Some reviewers pointed out The norm of DIV larger than DID proved in the paper does not convincingly demonstrate that it can make DDRep include less information and DIRep with more information, and further improve the performance of MaxDIRep with a larger norm of DIRep. I further checked the author's rebuttal and am **leaning to agree** with this concern. Author simply discussed the difference between DSN and their work. However, the reviewer's intention is to understand the general scenarios rather than DSN. This makes the supporting evidence a bit weak.

Based on the aforementioned discussions, I agree with reviewers' concerns and think a major revision is required for resubmission.

**Audience:**

Yes

**Claims And Evidence:**

NO. See the comments.

**Resubmission Of Major Revision:**

The authors may consider submitting a major revision at a later time.